# AOEPT: Breaking the Implicit Modality-Reduction Bottleneck in Modality-Missing Prompt Tuning

Jian Lang [1]  Rongpei Hong [1]  Ting Zhong [1]  Fan Zhou [1 2]

## Abstract

Deploying multimodal systems in real-world environments often entails handling modality-missing scenarios, where one or more modalities are unavailable. While recent studies address this challenge for the general Multimodal Transformer (MT) architecture via prompt tuning, we identify a fundamental limitation in these methods: the *Implicit Modality-Reduction bottleneck*. By conditioning prompts solely on the observed modalities, they inadvertently restrict the reasoning scope of MTs to the modality-reduced subspace, *cutting off* access to the latent information sources of the missing modalities. To overcome this limitation, we propose **AOEPT**, which pioneers a novel *modal-contextualized prompting* fashion. Specifically, we introduce lightweight *Modal-Contextualized Prompts (MCPs)* that distill global modality-wise priors from training data, serving as latent repositories of the information sources for missing modalities. Conditioned on the remaining modalities, these MCPs are instantiated into instance-aware prompts that selectively augment missing-modality information for each sample, thereby restoring the reasoning scope of MTs beyond the observed-modality-only subspace. Experiments across various multimodal benchmarks and backbones confirm the strong performance of AOEPT, with minimal computational overhead.

## 1. Introduction

Multimodal learning, which mimics the way humans perceive and understand the real world through the integration of heterogeneous information sources, such as visual, lin-

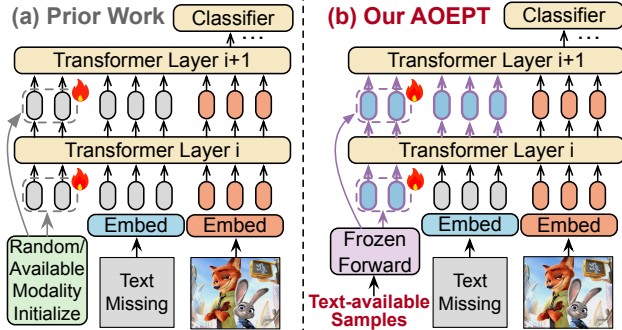

*Figure 1.* Paradigm comparison for an image-only sample between (a) Prior Work, which falls into the unimodal prediction bottleneck, and (b) Our AOEPT, which explicitly breaks such bottleneck.

guistic, and acoustic signals, has emerged as a central research problem (Yuan et al., 2025). Existing methods often assume the data in both training and deployment phases is modality-complete. However, in real-world scenarios, multimodal systems often operate under in-the-wild, dynamic, and noisy conditions (Lang et al., 2026; Hong et al., 2025; Li et al., 2026), where extreme situations such as sensor failures and transmission errors can render certain modalities unavailable, severely degrading their practical utility (Ma et al., 2021; Li et al., 2025). Consequently, developing robust systems that can maintain reliability under modality-missing scenarios is of critical importance for practicality.

Traditional modality missing learning methods, including unified multimodal learning approaches (Zhao et al., 2021) and modality imputation models (Cai et al., 2018; Ma et al., 2021), heavily rely on customized model architectures to handle missing modalities, which limits their generalizability and flexibility across a wide range of multimodal tasks (Xu et al., 2023). *Recently*, Multimodal Transformers (MTs), which adopt a unified and general architecture in processing multimodal data, have become a dominant choice for a wide range of applications (e.g., visual question answering (Marouf et al., 2025), Multimodal Large Language Model (MLLM) (Bai et al., 2025)). As a result, addressing modality-missing problems in MTs has attracted increasing attention from recent works (Ma et al., 2022; Zhao et al., 2025). Current approaches often adopt parameter-efficient prompt tuning strategies (Jia et al., 2022), where only a set of learnable prompts is employed to adapt the frozen pretrained

[1]University of Electronic Science and Technology of China, Chengdu, Sichuan [2]Intelligent Digital Media Technology Key Laboratory of Sichuan Province, Chengdu, Sichuan. Correspondence to: Ting Zhong <zhongting@uestc.edu.cn>.

*Proceedings of the 43rd International Conference on Machine Learning*, Seoul, South Korea. PMLR 306, 2026. Copyright 2026 by the author(s).

MTs to the incomplete multimodal inputs. MAPs (Lee et al., 2023) pioneered the missing-aware prompts for MTs in tackling incomplete samples. Subsequent studies, such as DCP (Hu et al., 2024) and MemPrompt (Zhao et al., 2025), further refined prompt design along various perspectives, such as sample-specific prompts and cross-modal shared prompts, leading to progressively improved performance. Then, a natural question arises: *do existing prompt-tuning methods fully tap into the potential of prompts for addressing modality-missing challenges in MTs?*

As illustrated in Figure 1 (a), we provide a **critical rethinking** of the paradigms in these prompting methods: Their prompts are often either *randomly initialized* (e.g., MAPs, MemPrompt) or *initialized using remaining available modalities* in incomplete samples to become sample-specific (e.g., DCP). Consequently, they can be regarded as merely treating prompts as learnable signals to fine-tune MTs for accommodating *degraded* and *modality-reduced* input structures, followed by a direct mapping from such incomplete observations to labels. *Worse still*, when a dual-modal sample suffers from missing modalities, these methods force MTs to reason solely within the unimodal space, therefore degrading a multimodal problem into a unimodal one, falling into the following bottleneck:

> **Implicit Modality-Reduction (IMR) Bottleneck:** Existing prompt tuning mechanism inadvertently constrains the reasoning scope of MTs to the modality-reduced subspace, failing to fully trigger the strong multimodal modeling capacity of MTs learned during pretraining.

To understand and alleviate this bottleneck, we conduct a very simple pilot experiment (cf. Section 4.2), where the randomly initialized prompts for text- or image-missing samples in baseline MAPs are instead initialized using the global text or image information from training samples. And we observe a performance improvement.

In light of these observations, we propose **AOEPT**, a novel missing-adaptive mod**A**l-c**O**nt**E**xtualized prom**PT**ing framework that shifts the paradigm from adapting MTs to the degradation to active compensation. As illustrated in Figure 1, AOEPT overcomes the Implicit Modality-Reduction bottleneck with an effective albeit minimalist prompting fashion. Specifically, AOEPT first forwards the training samples (including both complete and modality-missing ones) through the frozen MTs, and reorganizes the resulting layer-wise token representations into modality-specific information collections. Subsequently, a set of lightweight Modal-Contextualized Prompts (**MCPs**) is introduced to condense and distill the corresponding modality information from these collections. As a result, the MCPs serve as *modality-level latent repositories* that depict the global contextual information and distribution for each modality.

When handling incomplete samples, AOEPT adaptively fetches MCPs considering the specific missing patterns (e.g., image missing) in different samples, and instantiates them into **instance-aware prompts** conditioned on the remaining observed modalities. This instantiation process projects the modality-level representations into instance-specific space, selectively activating modality information most relevant to the current sample, and is further refined through a intra-modal latent consistency regularization. Finally, these prompts are inserted into the MTs to explicitly supplement the missing-modality information for each sample, effectively surmounting the Implicit Modality-Reduction Bottleneck. The contributions of this study are as follows:

- We revisit existing modality-missing prompt-tuning methods and identify the **Implicit Modality-Reduction (IMR)** bottleneck: they unintentionally confine the reasoning scope of MTs to modality-reduced subspace, cutting off access to latent information sources of missing modalities.

- We propose a conceptually novel solution **AOEPT**. It explicitly restores access to the information repositories of missing modalities via an efficient modal-contextualized prompting fashion, expanding the reasoning scope of MTs beyond that constituted by the remaining modalities.

- Experiments on diverse benchmarks show the efficacy of AOEPT. Moreover, we introduce a new metric, namely Normalized Missing-modality Mutual Information (**$NM^2I$**), to diagnose the severity of the IMR bottleneck. Furthermore, we empirically reveal a **modality information scaling bottleneck**, where performance of existing methods plateaus even as training conditions improve with more available information from the modality that missing at test time, while AOEPT can benefit from such additional information. Code is in https://github.com/Jian-Lang/AOEPT.

## 2. Related Work: Modality Missing Learning

Modality missing learning focuses on developing models that are robust to incomplete multimodal data encountered during deployment (Ma et al., 2021; Wu et al., 2024). Early studies can be broadly divided into two categories: (1) Unified multimodal learning methods (Wang et al., 2023a; Zhao et al., 2021), which learn shared multimodal representations and leverage these shared representations to handle incomplete inputs, (2) Modality imputation methods (Cai et al., 2018; Ma et al., 2021), which attempt to generate missing modalities from the remaining ones using sophisticated cross-modal reconstruction networks. Despite their effectiveness, these methods rely heavily on architecture-specific model designs to address modality-missing issues, which limits their applicability across a wide range of multimodal downstream tasks (Xu et al., 2023). Recently, with the prevalence of Multimodal Transformer (MT) as a general

architecture across diverse multimodal tasks, many studies have been devoted to enhancing the robustness of MTs under modality-missing scenarios (Ma et al., 2022; Lee et al., 2023; Hu et al., 2024). They develop various prompt tuning strategies (Jia et al., 2022) to efficiently fine-tune the MTs in handling the incomplete multimodal data. MAPs (Lee et al., 2023) was the first work to employ missing-aware prompts in tuning MTs to adapt to the missing modalities. Subsequently, MSPs (Jang et al., 2024) reduced the number of prompts in MAPs to modality-wise ones, while DCP (Hu et al., 2024), MemPrompt (Zhao et al., 2025), and SyP (Zhang et al., 2025) refined the prompts to be sample-aware, memory-driven, and cross-modality shared for improved robustness. Nevertheless, these methods can be regarded as simply leveraging prompts to signal MTs in adapting to the degraded multimodal input structures, which fall into the bottleneck of Implicit Modality-Reduction (IMR).

Although retrieval-based prompt-tuning studies such as RAGPT (Lang et al., 2025a) and REDEEM (Lang et al., 2025b) attempt to inject external multimodal evidence into prompts or reconstruct missing modalities, they do not identify the IMR bottleneck inherent in current prompt-tuning paradigms. As a result, their solutions rely on *external retrieval and reconstruction modules*, rather than maintaining standard lightweight prompt-tuning paradigm, leading to substantial training and inference overhead. Moreover, this dependence on external static retrieval may introduce high variance in sample-wise evidence during training, and make the compensated multimodal information vulnerable to noisy or mismatched retrieved instances at inference time. In contrast, AOEPT realizes an **implicit and internalized** self-retrieval mechanism, where global modality-wise contextual knowledge is first distilled into lightweight MCPs and then projected into instance-specific prompts conditioned on the observed modalities, realizing an efficient and noise-resilient prompt-tuning paradigm.

## 3. Methodology

### 3.1. Preliminary

**Rethinking of Existing Prompting Methods.** To simplify the formulation without loss of generality, we consider a dual-modal multimodal task, where each data $x$ contains text and image modalities $t$ and $v$. The dataset $\mathcal{D}$ contains three types of data, where $x = (t, v)$ is the modality-complete one, and $x = (t, \_)$ and $x = (v, \_)$ denote the text-only and image-only data. For clarity, we consider the single-stream MT, $F_\theta(\cdot)$, which can be simplified as a stack of $L$ transformer encoder layers: $F_\theta(\cdot) = f_\theta^L \circ f_\theta^{L-1} \circ \cdots \circ f_\theta^1(\cdot)$, with each layer $f_\theta^i(\cdot)$ taking the concatenation of multimodal information as input and performs self-attention.[1] Existing

[1]The dual-stream MT implementation is in Appendix A.

methods incorporate a set of learnable prompts into the frozen encoder layers of MT and optimize the prompts to enhance modality-missing robustness of the MT:

$$\arg \min_{C_\phi, \mathcal{G}_\psi} \; \mathbb{E}_{(x,y)\sim\mathcal{D}} \; [L(C_\phi(F_\theta(x; \mathcal{G}_\psi(z))), y)], \quad (1)$$

where $C_\phi(\cdot)$ is the task-specific classification head, $L(\cdot)$ is the task objective (e.g., Cross-Entropy $L_{\text{CE}}$). $\mathcal{G}_\psi(\cdot)$ is the prompt construction function, which takes a conditional signal $z$ to drive the corresponding prompts generation, and can be used as a unifying formulation for existing prompting methods. However, these methods often either randomly initialize prompts, where the signal $z$ can be ignored or reduced to coarse input-structure indicators (e.g. image-only structure) (Lee et al., 2023; Jang et al., 2024), or generate sample-specific prompts by using the available modalities in incomplete samples, i.e., $z \triangleq (t, \_)$ or $z \triangleq (v, \_)$ (Hu et al., 2024; Zhao et al., 2025). As a result, they can be cast as leveraging $\mathcal{G}_\psi(z)$ to adapt MTs to degraded and incomplete input structures, and MT's reasoning scope on modality-missing samples is inherently bounded to the subspace of the remaining modalities (which we refer to as Implicit Modality-Reduction (IMR) bottleneck).

**Workflow of AOEPT.** To overcome the IMR bottleneck, we propose AOEPT. AOEPT explicitly and adaptively augments the MTs with missing-modality information through a novel and lightweight modal-contextualized prompting fashion. Specifically, a set of Modal-Contextualized Prompts (MCPs) is constructed to distill the global modality-level contextual information from the training set (**Section 3.2**). Subsequently, these prompts are instantiated into instance-aware ones by conditioning on the remaining observed modalities, activating the information most relevant to the missing modalities for each data instance (**Section 3.3**). Finally, the resulting prompts are adaptively inserted into MTs for prompt tuning, breaking the confinement of the modality-reduced subspace and overcoming the IMR problem (**Section 3.4**). The workflow of AOEPT is in Figure 2.

### 3.2. Modal-Contextualized Prompt Construction

To alleviate the IMR bottleneck in existing methods, we empirically observe that, replacing randomly initialized prompts with text or image information from the training set as "informative priors" leads to clear performance improvements for MTs under text or image-missing scenarios (cf. Section 4.2). Inspired by this, we propose a set of Modal-Contextualized Prompts (**MCPs**), which distill the modality-specific global contextual information and distribution from the training set. Specifically, taking the Text-Contextualized Prompts (**TCPs**) construction as an example, we first feed the $N_t$ text-available training samples (i.e., both modality-complete and text-only ones) into the $L$ frozen MT encoder layers, and the resulting inferred layer-wise tokens form the

**Figure 2.** Workflow of AOEPT. (a) The TCPs are constructed from layer-wise inferred text-modal collections obtained via frozen forward passes on text-available samples through the MTs. (b) The TCPs are then projected into instance-aware ones conditioned on the remaining modalities, activating sample-specific informative cues associated with the missing modality for the MTs via the prompt tuning.

text-specific information collections:

$$\mathbf{C}_t^l = \{\mathbf{t}_1^l, \mathbf{t}_2^l, \cdots, \mathbf{t}_{N_t}^l\}, \ \mathbf{t}_{i,\_}^l = \text{Pool}(F_\theta^l(x_i)), \quad (2)$$

where $\mathbf{C}_t^l$ is the text-specific information collection derived from the $l$-th encoder layer, $l \in [0, L-1]$, with each element $\mathbf{t}_i^l \in \mathbb{R}^d$ is the sequence-pooled text token representation of each text-available sample $x_i$, $F_\theta^l(\cdot) = f_\theta^l \circ \cdots \circ f_\theta^1(\cdot)$, $d$ is the feature dimension, $\text{Pool}(\cdot)$ is the average pooling operation, and _ represents that image modality is ignored in this process. When $l = 0$, each $\mathbf{t}_i^0$ is derived from the embedding layer of MT. Nevertheless, the number of text-available samples $N_t$ can still be prohibitively large. To further reduce the collection size for efficiency, inspired by (Zhang et al., 2022), we group the token representations from collection $\mathbf{C}_t^l$ into $N_t'$ semantic prototypes with K-means clustering that capture fine-grained text-level distributions:

$$\arg\min_{S_i} \sum_{i=1}^{N_t'} \sum_{\mathbf{t}_j^l \in S_i} \|\mathbf{t}_j^l - \hat{\mathbf{t}}_i^l\|^2, \ \hat{\mathbf{t}}_i^l = \frac{1}{|S_i|} \sum_{\mathbf{t}_j^l \in S_i} \mathbf{t}_j^l, \quad (3)$$

where $\hat{\mathbf{t}}_i^l$ denotes $i$-th refined token representation (prototype) and $S_i$ is $i$-th cluster set. The refined collection is then formalized as $\hat{\mathbf{C}}_t^l = \{\hat{\mathbf{t}}_1^l, \hat{\mathbf{t}}_2^l, \cdots, \hat{\mathbf{t}}_{N_t'}^l\}$, where $N_t'$ satisfies $N_t' \ll N_t$. Subsequently, we propose three construction methods of TCPs in distilling the global text-specific contextual information from the collections. To simplify the following discussion, we focus on the $l$-layer prompts construction, where $l \in [1, L]$, and we define $n$: $n = l - 1$.

**Attention-based Construction Method.** We first randomly initialize a set of $M$ learnable prompts $\mathbf{P}^l = \{\mathbf{P}_1^l, \ldots, \mathbf{P}_M^l\} \in \mathbb{R}^{M \times d}$. We then leverage $\mathbf{P}^l$ as the query to condense the text-specific information from $\mathbf{C}_t^n$ via a cross-attention operation with a residual connection:

$$\mathbf{P}_{\text{TCP}}^l = \text{Attn}(\mathbf{P}^l, [\hat{\mathbf{t}}_1^n, \cdots, \hat{\mathbf{t}}_{N_t'}^n], [\hat{\mathbf{t}}_1^n, \cdots, \hat{\mathbf{t}}_{N_t'}^n]) + \mathbf{P}^l, \ (4)$$

$$\text{Attn}(\mathbf{Q}, \mathbf{K}, \mathbf{V}) = \text{Softmax}(\frac{\mathbf{Q}\mathbf{K}^\top}{\sqrt{d}})\mathbf{V}, \quad (5)$$

where $\mathbf{P}_{\text{TCP}}^l \in \mathbb{R}^{M \times d}$ denotes the $l$-layer TCP with length $M$, and $[,]$ is the concatenation operation. In addition, we further introduce two alternative construction methods with more or less computational overhead for MCP construction.

**MLP-based Construction Method.** We apply a Multi-Layer Perceptron (MLP) to the text collection, followed by an adaptive pooling (Guo et al., 2025) to form the TCPs:

$$\mathbf{P}_{\text{TCP}}^l = \Phi_{\text{pooling}}^{(M)}\Big(\text{MLP}\big([\hat{\mathbf{t}}_1^n, \cdots, \hat{\mathbf{t}}_{N_t'}^n]\big)\Big), \quad (6)$$

where $\mathbf{P}_{\text{TCP}}^l \in \mathbb{R}^{M \times d}$, $\Phi_{\text{pooling}}^{(M)}(\cdot)$ denotes a non-overlapping sliding-window based adaptive pooling operator that aggregates the input sequence into $M$ output tokens, and $\text{MLP} : \mathbb{R}^d \to \mathbb{R}^d$ represents the MLP.

**Initialization-based Construction Method.** To further reduce runtime cost, we directly apply adaptive pooling to the refined collections and use the pooled representations as the initialization (starting point) of the learnable TCPs:

$$\mathbf{P}_{\text{TCP}}^l(0) := \Phi_{\text{pooling}}^{(M)}([\hat{\mathbf{t}}_1^{l-1}, \ldots, \hat{\mathbf{t}}_{N_t'}^{l-1}]), \quad (7)$$

Here, $\mathbf{P}_{\text{TCP}}^l(0)$ denotes the prompt tokens for layer $l$ at initialization, which are treated as learnable parameters optimized via gradient descent during tuning.

**Discussion.** Compared to the attention-based construction, the MLP-based method introduces non-linear transformations over the refined modality collection, offering an *expressive* but more costly alternative. whereas the Initialization-based method achieves the most lightweight design. We adopt the Attention-based construction as the default, while providing an empirical evaluation on the efficiency and performance of three construction methods in Section 4.6.

At this stage, the TCPs act as a *latent text-specific repository* that can provide MT with global contextual information of the text modality, therefore restoring MT's reasoning scope from the image-only subspace and alleviating Implicit Modality-Reduction bottleneck caused by text missing.

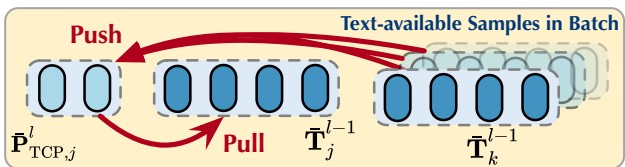

*Figure 3.* The intra-modal latent consistency regularization.

### 3.3. Instance-Aware Prompt Instantiation

After deriving the MCPs, a natural approach is to feed these prompts into MTs for incomplete inputs to complement missing-modality information. However, since MCPs capture the global, modality-level distribution, they are required to be further refined to adapt to each sample. Specifically, for an image-only sample $x_i = (v_i, \_)$, we leverage its remaining modality, $v_i$, as the condition to perform the instance-aware prompt instantiation, **selectively activating** modality-specific information stored in the TCPs that is most relevant for each sample $x_i$:

$$\mathbf{P}^l_{\text{TCP},i} \triangleq \mathcal{I}(\mathbf{P}^l_{\text{TCP}} \mid v_i) = \mathbf{P}^l_{\text{TCP}} \odot \sigma\big(\text{MLP}(\bar{\mathbf{V}}^{l-1}_i)\big), \quad (8)$$

where $\mathbf{P}^l_{\text{TCP},i}$ represents the instantiated, instance-aware TCP for $x_i$, $\bar{\mathbf{V}}^{l-1}_i \in \mathbb{R}^d$ is the sequence-pooled image hidden representations of sample $x_i$ yielded from encoder layer-$(l-1)$ (when $l = 1$, the $\bar{\mathbf{V}}^0_i$ is from the embedding layer), $\sigma$ is sigmoid function, $\odot$ is the element-wise product.

To further filter out the relevant text-modal information for each sample in a fine-grained manner, we introduce an **intra-modal latent consistency regularization** constraint (Figure 3) applied only to *text-available* training sample $x_j$:

$$L_{\text{CR}} = -\log \frac{\exp(\text{sim}(\bar{\mathbf{P}}^l_{\text{TCP},j}, \bar{\mathbf{T}}^{l-1}_j)/\tau)}{\sum_{k=1}^B \exp(\text{sim}(\bar{\mathbf{P}}^l_{\text{TCP},j}, \bar{\mathbf{T}}^{l-1}_k)/\tau)}, \quad (9)$$

where $\bar{\mathbf{P}}^l_{\text{TCP},j} \in \mathbb{R}^d$ is the pooled instance-aware TCP for text-available sample $x_j$, $\bar{\mathbf{T}}^{l-1}_j \in \mathbb{R}^d$ is the pooled text representation from layer-$(l-1)$ for $x_j$, and $\bar{\mathbf{T}}^{l-1}_k \in \mathbb{R}^d$ is the text representation for sample $x_k$ in current batch. The detailed derivation of $L_{\text{CR}}$ is provided in Appendix B.

### 3.4. Missing-Adaptive Prompt Tuning

When handling an image-only sample $x_i$, we first adaptively fetch the corresponding layer-wise MCPs, TCPs $\mathbf{P}^l_{\text{TCP}}$ in this situation, and instantiate the TCPs into instance-aware ones $\mathbf{P}^l_{\text{TCP},i}$. We then perform the prompt tuning using these instance-aware prompts for $x_i$. Specifically, for the first $N$ encoder layers of MT, we perform the prompt tuning while dropping the prompts propagated from the prior layer:

$$\_, \mathbf{H}^l_i = f^l_\theta([\mathbf{P}^l_{\text{TCP},i}, \mathbf{H}^{l-1}_i]), \ l \in [1, N], \quad (10)$$

where $\mathbf{H}^l_i$ is the hidden representations from $l$-th MT layer, $\_$ indicates that the prompts from prior layers are discarded.

**Algorithm 1** Algorithm of AOEPT (TCP as an example).

**Input:** Frozen MT $F_\theta$ with $L$ layers $f^*_\theta$; training set $\mathcal{D}$.
**Output:** Prediction $\hat{y}_i$ for sample $x_i$.
1: Get text-specific token representations $\mathbf{C}^*_t$ from $F_\theta$ over text-available samples in $\mathcal{D}$, and apply clustering to obtain refined layer-wise collections $\mathbf{C}^*_t$ (Eq. (2) – (3)).
2: Construct layer-wise TCPs $\mathbf{P}^*_{\text{TCP}}$ from $\mathbf{C}^*_t$ using one of the prompt construction methods (Eq. (4) – (7)).
3: **for** $l = 1$ to $N$ **do**
4:     Derive instance-aware $\mathbf{P}^l_{\text{TCP},i}$ using modality $v_i$, and apply consistency regularization $L_{\text{CR}}$ (Eq. (8) – (9)).
5:     Insert $\mathbf{P}^l_{\text{TCP},i}$ into the MT encoder layer $f^l_\theta(\cdot)$ and perform prompt tuning (Eq.(10)).
6: **end for**
7: Apply prompt tuning from layers $N + 1$ to $L$ (Eq.(11)).
8: Get $\hat{y}_i = C_\phi(\mathbf{H}^L_i)$ and update $\mathbf{P}^*_{\text{TCP},i}$ via $L_{\text{CR}}$ and $L_{\text{CE}}$.

*Table 1.* Statistics of three multimodal benchmarks.

| Dataset | # Image | # Text | # Train | # Val | # Test | # Class |
|---|---|---|---|---|---|---|
| MM-IMDb | 25,959 | 25,959 | 15,552 | 2,608 | 7,799 | 23 |
| HateMemes | 10,000 | 10,000 | 8,500 | 500 | 1,500 | 2 |
| Food101 | 90,688 | 90,688 | 61,174 | 6,798 | 22,716 | 101 |

In the remaining layers, the prompts $\mathbf{P}^l_{\text{TCP},i}$ are no longer newly initialized for each layer. Instead, they are inherited from the previous layer and propagated to subsequent layers:

$$\mathbf{P}^{l+1}_{\text{TCP},i}, \mathbf{H}^l_i = f^l_\theta([\mathbf{P}^l_{\text{TCP},i}, \mathbf{H}^{l-1}_i]), \ l \in [N+1, L]. \quad (11)$$

Finally, the last layer hidden representation of $x_i$, $\mathbf{H}^L_i$, is input into the classifier $C_\phi(\cdot)$ (e.g., an MLP), to derive the final prediction: $\hat{y}_i = C_\phi(\mathbf{H}^L_i)$. During training, only the MCPs and the classification head in AOEPT are tuned, using both the $L_{\text{CR}}$ and the $L_{\text{CE}}$. Algorithm of AOEPT is in Algorithm 1. Notably, AOEPT is readily extended to scenarios with more modalities situations, incurring only linear overhead with more modalities (cf. Appendix C). Although AOEPT is conceptually new and different from existing prompting methods, it introduces *no additional training data assumptions* beyond them. The efficiency and complexity analysis of AOEPT are in Section 4.9 and Appendix D, and a mathematical analysis of IMR is in Appendix E.

## 4. Experiments

### 4.1. Experimental Setup

We provide a brief experimental setup, with details in Appendix F, and additional experiment results in Appendix G.

**Benchmarks.** Following (Lee et al., 2023), in the main paper, we adopt three benchmarks (cf. Table 1): ❶ **MM-IMDb** (Arevalo et al., 2017): a multi-label benchmark for movie genre classification with both image and text modalities. We report F1-Macro (F1-M) as metric. ❷ **Hate-Memes** (Kiela et al., 2020): a hateful meme classification

*Table 2.* Performance (%) of prompt tuning baselines and AOEPT on three datasets under 70% and 90% missing rates across diverse missing scenarios. The best results are in **bold** and the second are underlined. LB denotes the (lower-bound) performance of MT.

| Methods | Venue | MM-IMDb | | | | HateMemes | | | | Food101 | | | |
| | | Text | Image | Both | Avg. | Text | Image | Both | Avg. | Text | Image | Both | Avg. |
| | | F1-M | F1-M | F1-M | F1-M | AUC | AUC | AUC | AUC | ACC | ACC | ACC | ACC |
| LB (**CLIP, Missing Rate 70%**) | N/A | 47.22 | 51.32 | 49.53 | 49.36 | 62.70 | 62.39 | 62.53 | 62.54 | 74.12 | 84.79 | 78.87 | 79.26 |
| MAPs (Lee et al., 2023) | CVPR'23 | 49.17 | 51.82 | 50.09 | 50.36 | 61.12 | 63.24 | 65.04 | 63.13 | 76.52 | 85.64 | 79.12 | 80.43 |
| DCP (Hu et al., 2024) | NeurIPS'24 | 49.99 | 52.77 | 50.70 | 51.15 | 62.82 | 64.12 | 66.08 | 64.34 | 78.87 | 87.32 | 81.87 | 82.69 |
| RAGPT (Lang et al., 2025a) | AAAI'25 | 49.02 | 51.52 | 49.96 | 50.17 | 67.38 | 64.63 | 66.70 | 66.24 | 79.55 | 86.47 | 81.72 | 82.58 |
| MemPrompt (Zhao et al., 2025) | IJCAI'25 | 49.55 | 52.83 | 50.40 | 50.93 | 66.37 | 62.90 | 64.93 | 64.73 | 79.59 | 87.11 | 82.47 | 83.06 |
| SyP (Zhang et al., 2025) | ICCV'25 | 49.68 | 53.19 | 52.77 | 51.88 | 68.94 | 66.98 | 68.42 | 68.11 | 79.56 | 88.67 | 82.45 | 83.56 |
| **AOEPT** | Ours | **51.50** | **54.86** | **53.31** | **53.22** | **71.12** | **67.96** | **69.80** | **69.63** | **80.77** | **88.86** | **83.24** | **84.29** |
| LB (**CLIP, Missing Rate 90%**) | N/A | 45.66 | 49.28 | 46.02 | 46.99 | 68.38 | 65.71 | 64.78 | 66.29 | 67.22 | 82.12 | 72.13 | 73.82 |
| MAPs (Lee et al., 2023) | CVPR'23 | 48.44 | 50.15 | 47.08 | 48.56 | 57.21 | 61.52 | 63.34 | 60.69 | 73.16 | 82.14 | 76.58 | 77.29 |
| DCP (Hu et al., 2024) | NeurIPS'24 | 48.40 | 51.79 | 48.23 | 49.47 | 62.08 | 63.87 | 66.78 | 64.24 | 75.26 | 85.78 | 79.87 | 80.30 |
| RAGPT (Lang et al., 2025a) | AAAI'25 | 48.40 | 51.79 | 48.23 | 49.47 | 68.00 | 65.01 | 65.06 | 66.02 | 76.62 | 86.24 | 79.61 | 80.82 |
| MemPrompt (Zhao et al., 2025) | IJCAI'25 | 48.20 | 51.27 | 48.80 | 49.42 | 67.43 | 64.58 | 59.34 | 63.78 | 73.02 | 86.11 | 78.05 | 79.06 |
| SyP (Zhang et al., 2025) | ICCV'25 | 48.86 | 51.06 | 48.82 | 49.58 | 69.70 | 64.54 | **68.93** | 67.72 | 76.33 | 86.41 | 81.03 | 81.26 |
| **AOEPT** | Ours | **50.54** | **53.89** | **49.91** | **51.45** | **70.53** | **66.84** | 68.35 | **68.57** | **77.47** | **87.03** | **81.67** | **82.06** |

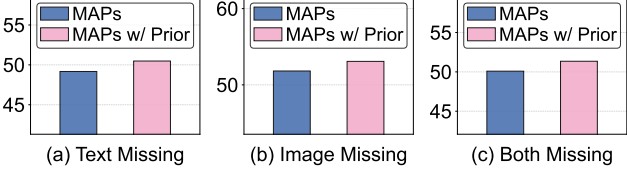

*Figure 4.* Performance of baseline MAPs without and with the missing-modality information priors on the MM-IMDb dataset.

task that leverages both image and text modalities. We use AUC as the metric. ❸ **Food101** (Wang et al., 2015): a 101-class food image–text classification task for recognition. We adopt Accuracy (ACC) as the metric.

**Baselines.** We adopt 5 competitive MT-oriented modality-missing baselines: MAPs (Lee et al., 2023), DCP (Hu et al., 2024), RAGPT (Lang et al., 2025a), MemPrompt (Zhao et al., 2025), and SyP (Zhang et al., 2025).

**Modality-Missing Protocol.** Following (Lee et al., 2023; Lang et al., 2025a), we adopt a more general and challenging modality-missing setting, where modality missing occurs in both training and test phases. We define the missing rate $\eta\%$ at both phases with three settings for dual-modal scenarios: ❶ **Text Missing** or ❷ **Image Missing** with rate $\eta\%$: $\eta\%$ of the samples are image-only or text-only, respectively, while the remaining $(100 - \eta)\%$ samples are complete. ❸ **Both Missing** with rate $\eta\%$: $\frac{\eta}{2}\%$ of the samples are text-only and $\frac{\eta}{2}\%$ are image-only, with the remaining $(100 - \eta)\%$ samples complete. We set $\eta\% = 70\%$ and $\eta\% = 90\%$ for the main evaluation, and also evaluate other missing rates.

**Implementation Details.** Following existing studies (Lee et al., 2023; Hu et al., 2024), we adopt the dual-stream MT, ❶ **CLIP ViT-B/16** (Radford et al., 2021), as the main backbone. Moreover, we also evaluate AOEPT on single-stream MT, ❷ **ViLT** (Kim et al., 2021), and a tri-modal MT, ❸ **MulT** (Tsai et al., 2019) in Appendix G.5. Refined collection capacity $N'_t$ is set to 256 for efficiency. The prompt length $M$ and prompt tuning depth $N$ are discussed in Section 4.6. All experiments use RTX 4090 GPUs.

### 4.2. Pilot Experiment: Unimodal Bottleneck

To understand and alleviate the Implicit Modality-Reduction (IMR) bottleneck in existing modality missing prompt tuning methods, we conduct a simple pilot experiment in a dual-modal situation on the MM-IMDb dataset. As illustrated in Figure 4, we simply replace the randomly initialized prompts in baseline MAPs (Lee et al., 2023) with prompts initialized using clustered text (w/ T Prior) or image (w/ I Prior) token representations for text- or image-missing samples, respectively. We observe performance improvements with these modified prompts, indicating that the original performance of MTs is bounded to the *degraded, single modality* input structure, despite their strong pretrained multimodal modeling capacity. And injecting the corresponding modality-contextual priors can mitigate this bottleneck.

### 4.3. Main Performance

We compare AOEPT with several MT-oriented modality-missing baselines, with results in Table 2. We observe that:

**(O1)** Existing prompting methods improve the modality-missing performance of MTs (LB). Methods such as MAPs, DCP, MemPrompt introduce a larger number of prompts with diverse types (e.g., sample-specific, memory-driven, synergistic static-dynamic) to refine the missing prompt tuning, while RAGPT employs instance-wise retrieval for missing-modality imputation and prompting.

*Table 3.* Ablation study of AOEPT under 70% text missing.

| | MM-IMDb | HateMemes | Food101 |
|---|---|---|---|
| **Variant** | F1-M | AUC | ACC |
| w/o MCP | 48.93 | 68.63 | 78.78 |
| w/o Instantiation | 49.17 | 69.42 | 79.13 |
| w/o Consistency | 50.56 | 69.85 | 79.59 |
| w/ Reconstruction | 48.55 | 70.13 | 76.81 |
| **AOEPT** | **51.50** | **71.12** | **80.77** |

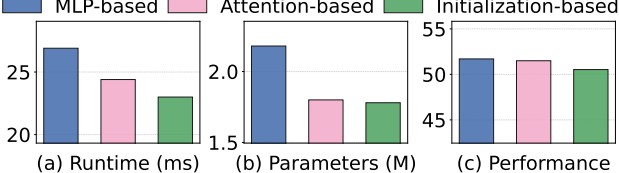

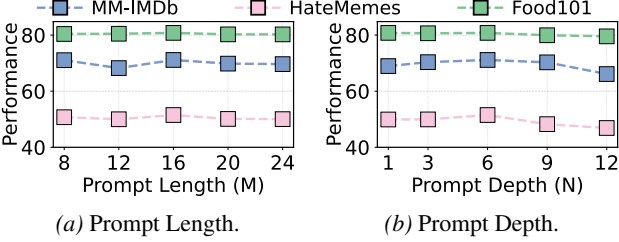

*Figure 5.* Comparison of three MCP construction methods in runtime costs, amount of learnable parameters, and performance.

*(a)* Prompt Length.      *(b)* Prompt Depth.

*Figure 6.* Performance of AOEPT with different prompt length and insertion positions under 70% text-missing case.

**(O2)** However, compared to the MAPs, the subsequent methods incur noticeable additional computational overhead (e.g., increased learnable parameters, instance-wise retrieval, memory mechanism). Moreover, most of them suffer from the Implicit Modality-Reduction (IMR) bottleneck, where the reasoning of the MT on incomplete samples is constrained by the degraded multimodal input structures. As a remedy, AOEPT effectively alleviates this bottleneck via an efficient solution, which explicitly replenishes sample-wise missing-modality information during lightweight modal-contextualized prompt tuning, achieving clear performance improvements with minimal learnable parameters.

### 4.4. Ablation Study

We analyze the role of core components within AOEPT and report the results in Table 3. Specifically, we design four variants: ① **w/o MCP**: the MCPs are replaced with vanilla randomly initialized prompts, like several baselines; ② **w/o Instantiation**: the MCPs are directly inserted into the MTs without instance-aware instantiation; ③ **w/o Consistency**: the remaining-modality consistency regularization is removed; ④ **w/ Reconstruction**: the MCPs are replaced by a modality-imputation network trained with a standard $L_2$ reconstruction loss, using a comparable number of learnable parameters to MCPs. We observe that **variant** ① incurs a clear performance drop, as the MTs are pushed back into

*Table 4.* Performance (%) of prompt tuning baselines and AOEPT on MM-IMDb under a 70% missing rate across diverse missing scenarios. The best results are in **bold** and the second are underlined. LB denotes the (lower-bound) performance of MT.

| | MM-IMDb | | | |
|---|---|---|---|---|
| | Text | Image | Both | Avg. |
| **Methods** | F1-M | F1-M | F1-M | F1-M |
| LB (**ViLT, Missing Rate 70%**) | 28.83 | 19.87 | 24.65 | 24.45 |
| MAPs (Lee et al., 2023) | 35.29 | 36.92 | 35.28 | 35.83 |
| DCP (Hu et al., 2024) | 34.15 | 38.18 | 35.86 | 36.06 |
| RAGPT (Lang et al., 2025a) | 36.19 | 39.90 | 36.74 | 37.61 |
| MemPrompt (Zhao et al., 2025) | 35.40 | 40.58 | 38.23 | 38.07 |
| SyP (Zhang et al., 2025) | 34.55 | 39.66 | 34.81 | 36.34 |
| **AOEPT** | **37.46** | **42.23** | **39.89** | **39.86** |

the unimodal bottleneck. Moreover, **variants ③ and ②** exhibit progressively degraded performance, underscoring the importance of selectively activating the most relevant information from the global modality-level repository for each sample. Finally, **variant ④** yields suboptimal performance, as a lightweight reconstruction network struggles to capture complex cross-modal mappings. Furthermore, the limited amount of modality-complete samples for reconstruction learning (i.e., 30%) further undermines its efficacy.

### 4.5. Performance on Single-Stream MT Backbone

Since AOEPT is model-agnostic and can be applied to various MT backbones, we further evaluate it on the single-stream MT, ViLT (Kim et al., 2021), with results in Table 4. We observe a conclusion similar to that of the main evaluation: AOEPT effectively alleviates the IMR bottleneck of the MT backbone and achieves the best performance.

### 4.6. In-Depth Analysis of Prompt Design

**Alternative Construction Methods Analysis.** We compare three MCP construction methods, and report the inference per batch runtime cost, amount of learnable parameters, and the performance on MM-IMDb dataset (F1-M) under 70% text-missing case in Figure 5. We observe that the MLP-based construction achieves slightly higher performance than the Attention-based one with additional computational overhead, whereas the Initialization-based method yields the lowest runtime cost but also the worst performance. Consequently, we adopt the Attention-based construction as the default choice, while the other two serve as alternatives for resource-constrained or resource-abundant settings.

**Prompt Length and Depth Analysis.** We evaluate the effectiveness of different prompt length $M$ and prompt tuning depth $N$ (i.e., the number of layers with newly instantiated MCPs). As illustrated in Figure 6, AOEPT initially benefits from longer prompts and deeper tuning depth, with performance peaking at $M=16$ and $N=6$. Consequently, we set $M=16$ and $N=6$ for an efficiency-performance trade-off.

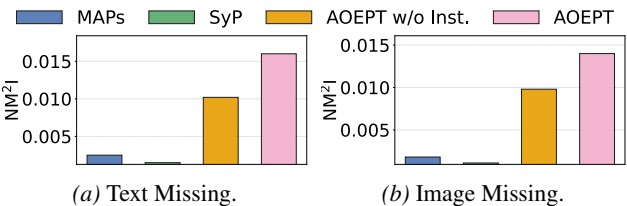

*(a)* Text Missing.    *(b)* Image Missing.

*Figure 7.* NM$^2$I comparison of AOEPT and baselines on the MM-IMDb dataset with 70% text- or image-missing cases.

### 4.7. NM$^2$I Analysis for Implicit Modality-Reduction

To further dissect whether AOEPT alleviates the IMR bottleneck by replenishing sample-specific missing-modality information for MTs via prompt tuning, we draw inspiration from Normalized Mutual Information (NMI) (Lancichinetti et al., 2009; Wang et al., 2024) and propose the metric **N**ormalized **M**issing-modality **M**utual **I**nformation (**NM$^2$I**). NM$^2$I quantifies how much information the prompt tokens share with the "ground-truth" latent representations of the missing modality at each MT layer, where the latter are obtained by forwarding that modality via the frozen MT.

Specifically, for each MT encoder layer $l$, we treat the prompt tokens tied to a certain modality-missing case as one random variable $\mathbf{P}_l$, and the latent representations of that modality, obtained from the same layer under the assumption that the modality is fully observed (modality complete), as another random variable $\mathbf{M}_l$. $\mathbf{M}_l$ is obtained by performing a frozen forward pass of the corresponding modality data from MT's layer $l$. We then model the relationship between $\mathbf{P}_l$ and $\mathbf{M}_l$ by approximating it with an empirical joint distribution $\tilde{e}_l(\mathbf{P}_l, \mathbf{M}_l)$, deriving by dot-product between their pair-wise token representations:

$$\tilde{e}_l(\mathbf{p}_l^k, \mathbf{m}_l^j) \triangleq \frac{\phi(\langle \mathbf{p}_l^k, \mathbf{m}_l^j \rangle)}{\sum_{j,k} \phi(\langle \mathbf{p}_l^k, \mathbf{m}_l^j \rangle)}, \quad (12)$$

where $\tilde{e}_l(\mathbf{p}_l^k, \mathbf{m}_l^j)$ represent the empirical joint probability of the prompt token $\mathbf{p}_l^k$ and the corresponding modality representation $\mathbf{m}_l^j$, respectively, reflecting their dependency at $l$-layer, $\phi(\cdot)$ denotes bounded, non-negative function (e.g., sigmoid), $\langle \cdot \rangle$ is the dot-product operation. Consequently, the marginal distributions of $\tilde{e}_l(\mathbf{P}_l)$ and $\tilde{e}_l(\mathbf{m}_l)$ are obtained through the marginalization operation:

$$\tilde{e}_l(\mathbf{p}_l^k) = \sum_j \tilde{e}_l(\mathbf{p}_l^k, \mathbf{m}_l^j), \;\; \tilde{e}_l(\mathbf{m}_l^j) = \sum_k \tilde{e}_l(\mathbf{p}_l^k, \mathbf{m}_l^j), \quad (13)$$

We then borrow the definition of the NMI (Lancichinetti et al., 2009) and calculate the NM$^2$I:

$$\text{NM}^2\text{I}_{(l)} = \frac{\text{MI}(\mathbf{P}_l; \mathbf{M}_l)}{\frac{1}{2}(\text{H}(\mathbf{P}_l) + \text{H}(\mathbf{M}_l))}, \quad (14)$$

where the mutual information $\text{MI}(\mathbf{P}_l; \mathbf{M}_l)$ and the entropies $\text{H}(\mathbf{P}_l)$ and $\text{H}(\mathbf{M}_l)$ are computed from the empirical joint

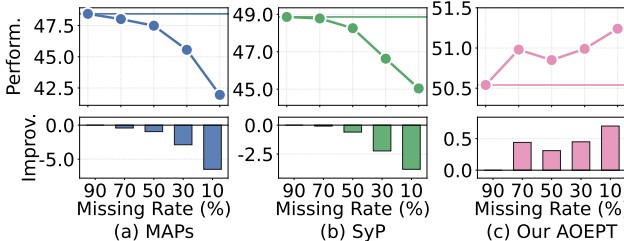

*(a)* MAPs  *(b)* SyP  *(c)* Our AOEPT

*Figure 8.* Performance of AOEPT and baseline methods under continually decreasing training modality-missing rate.

distribution and the corresponding marginal distributions, respectively. Concretely, $\text{H}(\mathbf{P}_l)$ can be computed as:

$$\text{H}(\mathbf{P}_l) = -\sum_k \tilde{e}_l(\mathbf{p}_l^k) \log \tilde{e}_l(\mathbf{p}_l^k), \quad (15)$$

and the mutual information term $\text{MI}(\mathbf{P}_l; \mathbf{M}_l)$ is given by:

$$\text{MI}(\mathbf{P}_l; \mathbf{M}_l) = \sum_{k,j} \tilde{e}_l(\mathbf{p}_l^k, \mathbf{m}_l^j) \log \frac{\tilde{e}_l(\mathbf{p}_l^k, \mathbf{m}_l^j)}{\tilde{e}_l(\mathbf{p}_l^k) \tilde{e}_l(\mathbf{m}_l^j)}. \quad (16)$$

Since NM$^2$I empirically quantifies the normalized mutual information between the prompt tokens and the latent representations of the missing modality, a higher value of NM$^2$I indicates that the prompts carry richer and more sample-specific information about the missing modality, thereby more effectively alleviating the IMR bottleneck.

As illustrated in Figure 7, we report the NM$^2$I values of AOEPT and baselines averaged across layers and test samples on the MM-IMDb dataset. We observe that baseline methods yield nearly zero NM$^2$I, which provides an alternative empirical perspective on the IMR bottleneck. In contrast, AOEPT effectively alleviates such bottleneck with clear NM$^2$I. Notably, without instance-aware projection, the prompts in variant AOEPT w/o Inst. (Instantiation) lack discriminability across samples, and provide limited informative missing-modality information at instance level.

### 4.8. Modality Information Scaling Bottleneck Analysis

In the main evaluation, we assume the same missing rate during training and testing. However, in real-world scenarios, modality-missing issues are more likely to occur at test time, while the training phase can often access more modality-complete data. Motivated by this practical consideration, we decrease the training text-missing rate from 90% to 10%, while fixing the test-time text-missing rate at 90%, on MM-IMDb dataset. Interestingly, in Figure 8(a)-(b), we observe that baseline prompting methods struggle to benefit from improved training conditions: their performance is hard to increase, but degrades as the training missing rate decreases, since training with lower missing rates makes them struggle to generalize to severely missing scenarios.

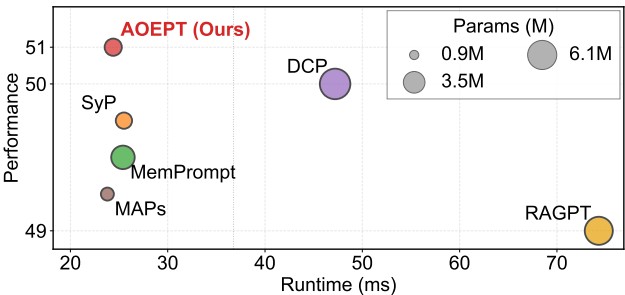

*Figure 9.* Efficiency comparison between AOEPT and baselines in terms of runtime costs and the number of learnable parameters.

Alternatively, maintaining the high training missing rate (i.e., 90%) causes the baseline performance to plateau (cf. horizontal lines in Figure 8(a)-(b)), a phenomenon we term the **Modality Information Scaling** bottleneck, which is a side effect of the Implicit Modality-Reduction problem. In contrast, in Figure 8(c), AOEPT benefits from improved training conditions, i.e., more information from the modality that is missing at test time. Notably, AOEPT does not reduce the missing rate (i.e., 90%) for model training. Nevertheless, its MCPs can leverage richer modality information (10% – 90%) to form more comprehensive global modal-contextual repositories, thereby leading to improved performance.

### 4.9. Efficiency Analysis

We compare the efficiency of AOEPT and baselines in Figure 9, with per-batch inference time and number of additionally introduced parameters, and performance on MM-IMDb under 70% text missing. We observe that AOEPT incurs comparable and even lower computational costs than baselines. The modest overhead mainly stems from the lightweight design of MCPs, which avoids costly components such as memory mechanisms or sample-wise retrieval (like MemPrompt, RAGPT), thereby achieving a favorable trade-off between efficacy and efficiency.

## 5. Conclusion

In this work, we proposed AOEPT, a conceptually novel framework that overcomes the Implicit Modality-Reduction (IMR) bottleneck in existing modality-missing prompt tuning methods. AOEPT alleviates such bottleneck through a lightweight yet principled modal-contextualized prompting strategy, effectively augmenting MTs with instance-aware missing-modality information. To quantify the IMR bottleneck, we further introduce Normalized Missing-modality Mutual Information ($NM^2I$) as a diagnostic metric, and leverage it to empirically validate the existence of this bottleneck in existing methods. Extensive experiments demonstrate the effectiveness of AOEPT, showing that it not only outperforms strong baselines but also achieves substantially higher $NM^2I$, with only modest computational overhead.

## Limitations

This study has following limitations and future directions.

First, $NM^2I$ serves as a diagnostic metric for the IMR bottleneck, but it is not necessarily monotonic with model performance. Specifically, in multimodal learning, strong modeling of the remaining modality may still achieve promising results in certain scenarios, even when the model remains confined to a modality-reduced reasoning scope. Nevertheless, for scenarios where the modality redundancy is limited, the impact of the IMR bottleneck becomes more pronounced, as the remaining modalities may not provide sufficient information for prediction. And for MTs pretrained with multimodal modeling capacity, restoring a sufficiently broad reasoning scope beyond the modality-reduced subspace remains crucial for robust modality-missing learning. Second, in AOEPT, we make a modest assumption commonly adopted in modality-missing learning: the semantic distributions of the training and test data do not differ significantly. Under this assumption, our MCPs can effectively distill modality-wise information from the training set to alleviate the IMR bottleneck for inference-time samples. Finally, unlike prior methods with randomly initialized, label-supervised prompts, AOEPT internalizes information distilled from the modality-reduced space through the efficient prompt tuning. Future work could explore additional constraints on the prompts to more effectively extract predictive information from the restored modality space.

## Acknowledgments

This work was supported by the National Natural Science Foundation of China (Grant No.62572097 and No. U23A20315). We would also like to thank a reviewer for remaining steadfast in assessment and providing insightful feedback that recognized the value of our work.

## Impact Statement

This paper identifies an inherent bottleneck, termed Implicit Modality-Reduction, in existing modality-missing prompt tuning methods for Multimodal Transformers (MTs). By revealing that current prompting mechanisms may unintentionally confine MTs to the modality-reduced subspace, this work provides a new perspective for understanding the limitations of existing approaches under incomplete multimodal inputs. To alleviate this IMR bottleneck, our method, AOEPT, restores the reasoning scope of MTs beyond the observed modalities by explicitly providing access to missing-modality contextual information. In this sense, it reframes modality-missing learning for MTs from passive adaptation to degraded input structures into an active information-access perspective. Additionally, this is achieved without introducing substantial computational overhead.

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

## A. Implementation of AOEPT on Dual-Stream Multimodal Transformer

In the main paper, we present the mathematical formulation of AOEPT on the single-stream MT for clarity. Nevertheless, extending AOEPT to dual-stream MTs is straightforward, as it follows the same formulation and differs only in the underlying MT architecture. Specifically, we formulate a dual-stream MT with text and image encoders, denoted as $F_\theta^t(\cdot)$ and $F_\theta^v(\cdot)$, respectively, potentially followed by a multimodal alignment module $M_\pi(\cdot)$. Each encoder can be simplified as a stack of L transformer encoder layers: $F_\theta^m(\cdot) = f_\theta^{m,L} \circ f_\theta^{m,L-1} \circ \cdots \circ f_\theta^{m,1}(\cdot)$, where $m \in \{t, v\}$. The prediction is given:

$$y = C_\phi\big(M_\pi\big(F_\theta^t(t), F_\theta^v(v)\big)\big), \tag{17}$$

where $C_\phi(\cdot)$ is the task-specific classification head, $t$ and $v$ are text and image modalities for each sample $x$. Subsequently, taking the Text-Contextualized Prompts (TCPs) construction as an example, we feed the text modality data from $N_t$ text-available training samples (i.e., both modality-complete and text-only ones) into the $L$ frozen MT text encoder layers, and the resulting inferred layer-wise tokens form the text-specific information collections:

$$\mathbf{C}_t^l = \{\mathbf{t}_1^l, \mathbf{t}_2^l, \cdots, \mathbf{t}_{N_t}^l\}, \qquad \mathbf{t}_i^l = \text{Pool}(F_\theta^{t,l}(t_i)), \tag{18}$$

where $t_i$ is the text modality data for each text-available sample $x_i$, $\mathbf{C}_t^l$ is the text-specific information collection derived from the $l$ text encoder layer, $l \in [0, L-1]$, with each element $\mathbf{t}_i^l \in \mathbb{R}^d$ is the sequence-pooled text token representation of sample $x_i$, $F_\theta^{t,l}(\cdot) = f_\theta^{t,l} \circ \cdots \circ f_\theta^{t,1}(\cdot)$, $d$ denotes the feature dimension, $\text{Pool}(\cdot)$ represents the average pooling operation. When $l = 0$, each $\mathbf{t}_i^0$ is derived from the text embedding layer of MT. The subsequent steps for TCPs construction and instance-aware instantiation follow the same manner as provided in the main paper (cf. Eq. (3) – (9)).

Subsequently, taking the image-only example $x_i$ as an example, for the first $N$ text encoder layers, we perform the missing-adaptive prompt tuning using the instance-aware TCPs, while dropping the prompts propagated from the prior layer:

$$\_, \mathbf{T}_i^l = f_\theta^{t,l}([\mathbf{P}_{\text{TCP},i}^l, \mathbf{T}_i^{l-1}]), \ l \in [1, N], \tag{19}$$

where $\mathbf{T}_i^l$ is the text hidden representations of sample $x_i$ from $l$-th MT layer, $\_$ indicates that the prompts from prior layers are discarded. Notably, when $l = 1$, $\mathbf{T}_i^0$ is simply padded with the embedding of an empty string for the text-missing sample. In the remaining layers, the prompts $\mathbf{P}_{\text{TCP},i}^l$ are no longer newly initialized for each layer. Instead, they are directly inherited from the previous layer and propagated to subsequent layers:

$$\mathbf{P}_{\text{TCP},i}^{l+1}, \mathbf{T}_i^l = f_\theta^{t,l}([\mathbf{P}_{\text{TCP},i}^l, \mathbf{T}_i^{l-1}]), \ l \in [N+1, L]. \tag{20}$$

Finally, the last-layer text representation of $x_i$, denoted as $\mathbf{T}_i^L$, together with the corresponding image representation $\mathbf{V}_i^L$, is fed into the multimodal fusion module $M_\pi$, and the fused representation is then passed to a task-specific classifier $C_\phi(\cdot)$ (e.g., an MLP) to produce the final prediction: $\hat{y}_i = C_\phi(M_\pi(\mathbf{T}_i^L, \mathbf{V}_i^L))$.

## B. Derivation of Intra-Modal Latent Consistency Regularization

In this section, we provide a detailed derivation of the proposed intra-modal latent consistency regularization constraint $L_{\text{CR}}$. Since the instance-aware TCPs are derived from the global ones, we design this constraint to further disentangle the most relevant information from the global text-modal repositories for each data instance. Taking the instance-aware TCP $\bar{\mathbf{P}}_{\text{TCP},i}^l$ for sample $x_i$ as an example, a more straightforward way is to leverage the latent representations of the remaining modality (i.e., image modality) for the consistency regularization, which can be formulated as follows:

$$L_{\text{CR}} = -\log \frac{\exp(\text{sim}(\bar{\mathbf{P}}_{\text{TCP},i}^l, \bar{\mathbf{V}}_i^{l-1})/\tau)}{\sum_{k=1}^B \exp(\text{sim}(\bar{\mathbf{P}}_{\text{TCP},i}^l, \bar{\mathbf{V}}_k^{l-1})/\tau)}, \tag{21}$$

where $\bar{\mathbf{P}}_{\text{TCP},i}^l \in \mathbb{R}^d$ is the sequence-pooled instance-aware TCP, $\bar{\mathbf{V}}_k^{l-1} \in \mathbb{R}^d$ is the pooled image representation from layer-$(l-1)$ of image-available sample $x_k$ in current batch. However, such a constraint may encourage the TCPs to collapse toward the remaining modality, which contradicts our design principle of overcoming the Implicit Modality Degradation bottleneck. Consequently, we formulate this constraint within the scope of intra-modality and propose the intra-modal latent consistency regularization (Eq. (9)), which performs the consistency regularization of the instance-aware TCPs in the text modality space. Notably, this regularization requires the **modality-complete samples** under dual-modality scenarios.

Specifically, following prior studies (Hu et al., 2024), we insert all types of MCPs for each sample without considering sample-specific modality-missing conditions (i.e., image-only, text-only or complete). Consequently, for a modality-complete sample where the consistency regularization can be applied, the regularization constraint $L_{\text{CR}}$ is simultaneously applied to optimize both type of MCPs (i.e., TCPs and Image-Contextualized Prompts (ICPs)), which leads to more effective supervision. Specifically, the mathematical formulation of $L_{\text{CR}}$ for a modality-complete sample $x_j$ is given by:

$$L_{\text{CR}} = -\log \frac{\exp(\text{sim}(\bar{\mathbf{P}}_{\text{TCP},j}^l, \bar{\mathbf{T}}_j^{l-1})/\tau)}{\sum_{k=1}^{B_t} \exp(\text{sim}(\bar{\mathbf{P}}_{\text{TCP},j}^l, \bar{\mathbf{T}}_k^{l-1})/\tau)} - \log \frac{\exp(\text{sim}(\bar{\mathbf{P}}_{\text{ICP},j}^l, \bar{\mathbf{V}}_j^{l-1})/\tau)}{\sum_{k=1}^{B_v} \exp(\text{sim}(\bar{\mathbf{P}}_{\text{ICP},j}^l, \bar{\mathbf{V}}_k^{l-1})/\tau)}, \tag{22}$$

where $B_t$ and $B_v$ are the number of text-available or image-available samples in the current batch.

## C. Extension of AOEPT to Multiple Modalities

In the main paper, we provide the derivation of AOEPT under dual-modal setting for clarity. In this section, we extend AOEPT to the general multi-modal setting. We consider a model with $K$ modalities, denoted as $\{m_k\}_{k=1}^K$, where $K \geq 2$. This extension does not require any modification to the backbone MT or the design of AOEPT, but only requires adapting the formulation of instance-aware prompt instantiation to the multiple modalities setting. Specifically, for instance-aware prompt instantiation, we take the MCP associated with missing-modality $m_k$ as an example. Let $\mathbf{C}_t \subseteq \{m_1, \ldots, m_K\} \setminus \{m_k\}$ denote the set of remaining observed modalities for a given sample $x_i$. The instance-aware prompt instantiation process is:

$$\mathbf{P}_{\text{MCP-k},i}^l \triangleq \mathcal{I}\big(\mathbf{P}_{\text{MCP-k}}^l \mid \mathbf{C}_t\big) = \mathbf{P}_{\text{MCP}_k}^l \odot \mathcal{A}\big(\{\sigma\big(\text{MLP}_j(\bar{\mathbf{m}}_{i,j}^{l-1})\big) \mid m_j \in \mathbf{C}_t\}\big), \tag{23}$$

where $\mathbf{P}_{\text{MCP-k}}^l$ is the $l$-layer MCP for modality $m_k$, the $\mathbf{P}_{\text{MCP-k},i}^l$ is the instance-aware MCP-k for sample $x_i$, $\bar{\mathbf{m}}_{i,j}^{l-1}$ is the $l-1$ layer sequence-pooled representation for modality $m_j$ of sample $x_i$. $\mathcal{A}(\cdot)$ is the aggregation function, with an example simple implementation using the average-based aggregation (employed in this study):

$$\mathcal{A}(\{\mathbf{E}_j \mid m_j \in \mathbf{C}_t\}) = \frac{1}{|\mathbf{C}_t|} \sum_{m_j \in \mathbf{C}_t} \mathbf{E}_j. \tag{24}$$

Since MCP is designed to be modality-wise, AOEPT is readily extended to multiple modalities scenarios, incurring only linear overhead with respect to the number of modalities.

## D. Complexity Analysis of AOEPT

In this section, we provide a complexity analysis of AOEPT. Specifically, we decompose the analysis into three stages: ① Offline modality collection construction and refinement, ② Modal-Contextualized Prompt (MCP) construction, and ③ Instance-aware prompt instantiation of MCP. Notably, we take the pipeline for the image-only samples as an example.

**Definition D.1.** Let $N_t$ denote the number of text-available training samples, $N_t'$ represent the number of refined text token representations (modality prototypes) in the collection after clustering ($N_t' \ll N_t$), $M$ is the number of prompt tokens (prompt length) per layer, $I$ is the number of clustering iterations, $d$ is the hidden dimension, $d$ denote the feature dimension, $l$ denote the feature sequence length, and $L$ is the total number of encoder layers in the MT.

### D.1. Offline Modal-Specific Information Collection Construction and Refinement

The offline component in AOEPT is the construction and refinement of modality-specific representation collections. Specifically, all text-available training samples are forwarded through the frozen MT to extract layer-wise pooled representations, forming the raw modality collection $\mathbf{C}_t^l$ at each layer. Each self-attention layer has a computational complexity of $\mathcal{O}(4ld^2 + 2l^2d)$, where the quadratic term $\mathcal{O}(l^2 \cdot d)$ dominates in practice and the original form is therefore simplified as $\mathcal{O}(l^2d)$. Consequently, this step incurs a cost of $\mathcal{O}(N_t \cdot L \cdot l^2d)$, where $L$ is usually a small positive constant in practical MT backbones, and it can be reduced to $\mathcal{O}(N_t \cdot l^2d)$. To further improve efficiency, we apply K-means clustering to refine the raw collections into $N_t'$ semantic prototypes per layer. The K-means refinement step incurs a computational cost of $\mathcal{O}(N_t \cdot N_t' \cdot d \cdot I)$, In practice, $I$ is a bounded constant. Moreover, this stage is only conducted once before the training, and incurs zero inference time overhead. We empirically observe that on the MM-IMDb dataset under 70% text missing, this state only costs about 3.4 minutes, which is equivalent to just adding a *single* training epoch (about 3.5 minutes).

## D.2. MCP Construction

In the following, we analyze the three MCP (TCP) construction strategies separately.

**Attention-based Construction**. In this method, $M$ learnable prompt tokens attend to the refined text-modal information collection via cross-attention. For each layer, the dominant cost arises from computing attention between $M$ queries and $N_t'$ keys, resulting in a complexity of $\mathcal{O}(M \cdot N_t' \cdot d)$. Across all L layers, the total MCP construction cost is $\mathcal{O}(L \cdot M \cdot N_t' \cdot d)$. This cost is independent with each sample, and does not scale with the number of samples processed.

**MLP-based Construction.** The MLP-based method applies a shared Multi-Layer Perceptron to each refined text token (prototype) followed by adaptive pooling. The dominant computation stems from the MLP transformation over $N_t'$ prototypes, yielding a per-layer cost of $\mathcal{O}(N_t' \cdot d^2)$, and a total cost of $\mathcal{O}(L \cdot N_t' \cdot d^2)$. Compared to the attention-based method, this strategy trades higher computational cost for stronger non-linear modeling capacity, since $d > M$ in practical.

**Initialization-based Construction**. The initialization-based method directly applies adaptive pooling over the refined text prototypes to obtain prompt initializations, without additional learnable transformations during construction. This results in a per-layer complexity of $\mathcal{O}(N_t' \cdot d)$, and a total cost of $\mathcal{O}(L \cdot N_t' \cdot d)$. This method is the most computationally lightweight among the three and serves as an efficient alternative when computational resources are limited.

Notably, when extending to multiple modalities, since the MCPs are designed in a modality-wise manner, the overall computational overhead scales linearly with the number of modalities $W$, i.e., $\mathcal{O}(W)$.

## D.3. Instance-aware Prompt Instantiation

Subsequently, MCPs are instantiated into instance-aware prompts conditioned on the remaining observed modalities and then used for prompt tuning. Compared to conventional prompt tuning, AOEPT introduces only a small amount of additional computation from the instance-aware instantiation step. Specifically, for each sample and each layer, instantiation consists of a lightweight MLP projection followed by element-wise modulation of the prompt tokens. The per-sample computational cost is dominated by the MLP projection, which scales as $\mathcal{O}(d^2)$, while the element-wise gating over $M$ prompt tokens incurs an additional $\mathcal{O}(M \cdot d)$ cost. Therefore, the total per-sample instantiation overhead is $\mathcal{O}(d^2 + M \cdot d)$.

# E. Rethinking Prompt Tuning for Modality Missing Learning via Information Theory

We provide an information-theoretic perspective to justify (i) why prompting mechanisms in existing methods are inherently restricted under the modality-reduced subspace (constituted using the remaining modalities), and (ii) how AOEPT alleviates this restriction (IMR bottleneck) by introducing an explicit information-access path to modality-specific contextual priors distilled from training data. For clarity, we analyze the inference-time information flow with the trained prompting mechanism fixed. Let $v$ and $t$ denote the observed image modality and missing text modality, respectively.

**Lemma E.1** (Information-Access Limitation of Observed-Only Prompting). *If a prompting mechanism generates prompts solely from the observed modality signal $z$ ($z \triangleq (t, \_)$ or $z \triangleq (v, \_)$ (Hu et al., 2024; Zhao et al., 2025)) and instance-independent noise $\varepsilon$ (e.g., random initialization (Lee et al., 2023; Jang et al., 2024)):*

$$\mathbf{P} = \mathcal{G}_\psi(z, \varepsilon), \quad where \ \varepsilon \perp (z, t), \tag{25}$$

$\mathcal{G}_\psi(\cdot)$ *is the prompt construction function, which takes the signal $z$ to drive the generation of prompts. Then the prompt* $\mathbf{P}$ *provides no instance-wise information sources of missing text modality beyond what is already contained in $z$:*

$$I(t; \mathbf{P} \mid z) = 0. \tag{26}$$

*Proof sketch.* Although the parameters $\psi$ in the prompt construction function may encode dataset-level statistics acquired from training data, at *inference time*, the instance-wise prompt $\mathbf{P}$ is generated solely from the observed modality signal $z$ (up to instance-independent noise $\varepsilon$). By construction, this induces the conditional independence $\mathbf{P} \perp t \mid z$ (equivalently, the Markov chain $t \to z \to \mathbf{P}$), since $\varepsilon$ is independent of $(z, t)$. Therefore, $I(t; \mathbf{P} \mid z) = 0$. This indicates that observed-modality-only prompting mechanism does not introduce an additional instance-wise information-access path to the information repositories of the missing modalities beyond the modality-reduced subspace. $\square$

Lemma E.1 shows a mechanistic limitation of observed-only prompting methods, where prompt generation process depends

solely on the observed signal $z$ (Implicit Modality-Reduction bottleneck). We now show that our AOEPT alleviates this limitation by introducing an additional conditioning variable $\mathbf{C}_t$.

**Proposition E.2** (AOEPT Establishes an Explicit Information-Access Path). *AOEPT generates prompts by conditioning on both the observed modality signal $z$ and a modality (text)-specific information repository $\mathbf{C}_t$ distilled from training data:*

$$\mathbf{P} = \mathcal{G}_\psi(z, \mathbf{C}_t). \tag{27}$$

*Under a mild non-degeneracy condition that the prompt generation function $\mathcal{G}_\psi$ does not ignore $\mathbf{C}_t$ given $z$, we have*

$$I(\mathbf{P}; \mathbf{C}_t \mid z) > 0, \tag{28}$$

*which indicates that the information-access path from prompts to modality-specific information repositories is established.*

*Proof sketch.* In existing methods (Lemma E.1), the prompt $\mathbf{P}$ is generated solely as a function of the observed signal $z$ and instance-independent noise $\varepsilon$, which implies that $\mathbf{P}$ is statistically independent of the modality-specific repository $\mathbf{C}_t$ given $z$. In contrast, AOEPT explicitly introduces $\mathbf{C}_t$ as a necessary conditioning variable in the prompt generation process, i.e., $\mathbf{P} = \mathcal{G}_\psi(z, \mathbf{C}_t)$. Under a mild non-degeneracy assumption that $\mathcal{G}_\psi$ does not ignore $\mathbf{C}_t$ given $z$ (which we empirically validate via NM$^2$I analysis), the generated prompt $\mathbf{P}$ necessarily depends on $\mathbf{C}_t$, leading to $I(\mathbf{P}; \mathbf{C}_t \mid z) > 0$. □

**Remark.** Proposition E.2 does *not* imply that AOEPT recovers the exact instance-level missing data $t$. Instead, it formalizes that AOEPT builds a valid *information-access path* via $\mathbf{C}_t$, enabling the MTs to access to the information repositories of the missing modalities beyond the observed, reduced modality subspace, alleviating the Implicit Modality-Reduction bottleneck.

# F. Detailed Experimental Setup

In this section, we provide detailed experimental setup, including the ① dataset descriptions, ② baseline descriptions and implementations, ③ MT backbone descriptions and implementations, and ④ the implementation details.

### F.1. Benchmarks

To fully evaluate the effectiveness of AOEPT, we compare it with four benchmarks. Specifically, following prior study (Lee et al., 2023), we first evaluate it on three dual-modal benchmarks: ❶ **MM-IMDb** (Arevalo et al., 2017), ❷ **HateMemes** (Kiela et al., 2020), and ❸ **Food101** (Wang et al., 2015). We also evaluate AOEPT on a tri-modal benchmark ❹ **IEMOCAP** (Busso et al., 2008) to showcase its effectiveness in extending to multiple modalities. Below, we present the dataset descriptions.

▷ **MM-IMDb** is a multimodal dataset designed for movie genre classification. It comprises two distinct modalities: visual (movie poster images) and textual (plot summaries). This dataset is primarily used for a multi-label classification, as each movie can be associated with multiple genres simultaneously. Following prior work (Lee et al., 2023; Hu et al., 2024), we adopt F1-Macro (F1-M) as metric.

▷ **HateMemes** focuses on identifying hate speech in memes via utilizing image and text modalities. To prevent the model from relying on a single modality, it is designed to make unimodal models more likely to fail by incorporating challenging samples known as "benign confounders", while simultaneously enhancing the performance of multimodal models. Following prior work (Lee et al., 2023), we adopt AUC as metric.

▷ **Food101** is a large-scale multimodal dataset designed for the multi-class classification task of food categories. This dataset uniquely pairs noisy image and text data across a diverse range of 101 food categories. Compiled using Google Image Search, it inherently incorporates real-world noise and variability, presenting both challenges and opportunities for robust model development in food recognition tasks. Following prior work (Lee et al., 2023), we adopt Accuracy (ACC) as metric.

▷ **IEMOCAP** is a widely used benchmark for speech emotion recognition and multimodal affective computing. It contains recorded videos from ten actors in five dyadic conversation sessions, and approximately 12 hours of data. Following previous works (Tsai et al., 2019; Wang et al., 2019), four emotions (happiness, anger, sadness and neutral state) are selected for emotion recognition, and we leverage the average accuracy (ACC) and F1-weighted score (F1) as evaluation metrics.

## F.2. Baseline Methods

In this study, we compare AOEPT with 5 competitive MT-oriented modality-missing baselines, including MAPs (Lee et al., 2023), DCP (Hu et al., 2024), RAGPT (Lang et al., 2025a), MemPrompt (Zhao et al., 2025), and SyP (Zhang et al., 2025). Following, we provide detailed descriptions for each baseline model.

▷ **MAPs** introduces missing-aware prompts that are strategically placed at various locations within MTs to address scenarios involving missing modalities. Specifically, it designs two types of prompt insertion strategies: attention and input level.

▷ **DCP** enhances missing-modality robustness by designing prompts that explicitly capture correlations between prompt signals and input features, as well as inter-layer prompt relationships. Specifically, DCP incorporates correlated, dynamic, and modal-common prompts that better leverage modality complementarity for varying missing cases.

▷ **RAGPT** introduces a retrieval-augmented prompt tuning framework where similar instances are retrieved to recover missing modality information and generate context-aware prompts, at the cost of additional instance-wise multimodal retrieval and reconstruction modules.

▷ **MemPrompt** introduces a memory-driven prompting framework to adaptively compensate for missing modalities. It uses a prompt memory storing modality-specific semantic information to retrieve semantically similar cues (generative prompts) and shared prompts to exploit cross-modal compensation from observed modalities.

▷ **SyP** employs a synergistic prompting strategy that jointly learns static and input-conditioned dynamic prompts via adaptive scaling, enabling more flexible adaptation to diverse missing patterns.

For the MM-IMDb dataset, we re-run all baseline methods instead of directly reporting the numbers from prior papers, with the reason: We observe an inconsistency in the public implementations, where individual movie plots are treated as separate samples while the missing rate is controlled at the movie level, resulting in a deviation between the specified and actual missing rates (e.g., a movie with multiple plots will be duplicated into several samples, while all duplicated samples sharing the same missing-modality label). To ensure a fair and controlled comparison, we reproduce all baseline results on MM-IMDb under a unified preprocessing pipeline, where each movie is treated as a single data instance to accurately control the missing rate, rather than treating individual plots as separate samples. This setting is also consistent with the original definition of the MM-IMDb dataset (Arevalo et al., 2017). For all baselines on other datasets, we report the results of these datasets directly from their original papers when available. We additionally reproduce the results for backbones that are not reported in the original papers using the official implementations.

In the main performance evaluation, we report a Lower Bound (**LB**) baseline to assess the inherent robustness of the MT backbones and to quantify the performance gains brought by prompt tuning methods under modality-missing scenarios. Specifically, the MT backbones are trained and evaluated under the same missing-rate and missing-type settings as all comparison methods. The *only difference* is that training is restricted to the trainable components of the MT backbone (as described in the next section) and the task-specific classifier, without introducing any learnable prompts. To ensure a fair comparison, the training protocol and hyper-parameters strictly follow those of MAPs (Lee et al., 2023).

## F.3. MT Backbones

In this study, to evaluate the scalability of our AOEPT, we first adopt two dual-modal MT backbones, including a double-stream MT ❶ **CLIP ViT-B/16** (Radford et al., 2021) and a single-stream MT ❷ **ViLT** (Kim et al., 2021). Moreover, we also adopt a tri-modal MT backbone ❸ **MulT** (Tsai et al., 2019) to showcase the effectiveness of AOEPT in extending to multiple modalities. Below, we provide a detailed implementation the backbones:

▷ **CLIP:** For CLIP, we adopt the pretrained ViT-B/16 variant following prior studies (Hu et al., 2024). During training, the complete CLIP model remains frozen while the modality-specific projection layer and final layer-norm are trainable parameters. Following prior work (Hu et al., 2024), the task-specific classifier consists of a single-layer MLP.

▷ **ViLT:** For ViLT, we adopt the pretrained model following existing studies (Lee et al., 2023). During training, the full ViLT model is frozen while the pooler layer remains trainable. Following prior studies (Lee et al., 2023), the task-specific classifier is implemented as a two-layer MLP.

▷ **MulT:** For MulT, we adopt the model architecture and pretrain the model on the MOSI (Zadeh et al., 2016) dataset. During pretraining, all parameters are trainable and optimized using Adam with learning rate $1 \times 10^{-3}$ for 40 epochs. Then for the target dataset IEMOCAP (Busso et al., 2008) (i.e., the one that we evaluate the modality-missing performance),

*Table 5.* Performance of AOEPT using different down-sampling strategies on three datasets under a 70% missing rate.

| | | MM-IMDb | | | HateMemes | | | Food101 | | |
| | | Text | Image | Both | Text | Image | Both | Text | Image | Both |
| Method | DS Strategy | F1-M | F1-M | F1-M | AUC | AUC | AUC | ACC | ACC | ACC |
|---|---|---|---|---|---|---|---|---|---|---|
| AOEPT | w/ Pooling | 50.67 | 53.64 | 50.16 | 70.23 | 66.53 | 68.94 | 79.66 | 87.46 | 82.64 |
| | w/ Clustering | **51.50** | **54.86** | **53.31** | **71.12** | **67.96** | **69.80** | **80.77** | **88.86** | **83.24** |

*Table 6.* Performance of AOEPT when scaling the capacity of the down-sampled modal-information collection on the MM-IMDb dataset.

| | Text Missing | | | | | | | | Image Missing | | | | | | | |
|---|---|---|---|---|---|---|---|---|---|---|---|---|---|---|---|---|
| $N_t'$ | 16 | 32 | 64 | 128 | 256 | 512 | 1024 | 2048 | 16 | 32 | 64 | 128 | 256 | 512 | 1024 | 2048 |
| **F1-M** | 50.40 | 50.10 | 50.20 | 50.00 | 51.50 | 51.10 | 50.30 | **51.70** | 54.50 | 54.40 | 53.80 | 54.20 | 54.86 | 55.06 | **55.26** | 55.16 |

the modality-specific projection layers are reinitialized to adapt to the dataset input dimensions. The classification head is implemented as a two-layer MLP with residual connections.

### F.4. Implementation Details

We set the refined collection capacity $N_t'$ is set to 256 for efficiency. The prompt length $M$ is selected from $\{8, 12, 16, 20, 24\}$ and tuning depth $N$ is selected from $\{1, 3, 6, 9, 12\}$. Clustering iteration number is 300. We train AOEPT using the AdamW optimizer (Loshchilov & Hutter, 2019) with a learning rate of $1 \times 10^{-2}$ and a weight decay of $2 \times 10^{-2}$ for 20 epochs. Following prior studies (Hu et al., 2024), we insert all type of MCPs into each sample. For the missing modalities, we follow prior studies (Lee et al., 2023). Specifically, we set the input text to an empty string for text-missing samples and set all pixel values to ones for image-missing samples. For the missing tables in all experiments, we randomly generate three fixed missing tables for each experimental combination of dataset, missing rate, and missing type. We evaluate our method and all baselines three times (once per missing table) and report the average performance across these three runs. Following prior work (Hu et al., 2024), we adopt bottleneck MLP for efficiency. All experiments are conducted on servers equipped with NVIDIA GeForce RTX 4090 GPUs.

## G. Additional Experimental Results

### G.1. Evaluation of Different Down-Sampling Strategies

In addition to the k-means clustering based down-sampling strategy adopted in the main paper for refining the original modal-specific information collections, we also explore an alternative, lightweight one: pooling-based down-sampling. Specifically, we formalize the pooling-based down-sampling strategy as follows:

$$\hat{\mathbf{t}}_i^l = \frac{1}{w} \sum_{k=(i-1)w+1}^{iw} \mathbf{t}_k^l, \quad i = 1, \ldots, N_t', \tag{29}$$

where $w$ is the window size, and $N_t' = \lfloor \frac{N_t}{w} \rfloor$, $\hat{\mathbf{t}}_i^l$ denotes $i$-th refined token representation. The refined collection is then formalized as $\hat{\mathbf{C}}_t^l = \{\hat{\mathbf{t}}_1^l, \hat{\mathbf{t}}_2^l, \cdots, \hat{\mathbf{t}}_{N_t'}^l\}$, where $N_t'$ satisfies $N_t' \ll N_t$. As shown in Table 5, we observe that the lightweight pooling-based down-sampling strategy leads to inferior performance. However, the performance degradation is not pronounced. Consequently, this alternative strategy remains a viable option in resource-constrained scenarios.

Moreover, in the main paper, we set the capacity of the refined modality-specific information set, $N_t'$, to 64 for efficiency considerations. In this place, we further analyze larger values of $N_t'$ by scaling the number of refined tokens in the collections, in order to explore whether increased capacity can lead to additional performance gains. As presented in Table 6, we empirically observe that increasing the collection capacity yields only marginal performance gains for AOEPT. Consequently, we set $N_t'$ to 256 to strike a balance between performance and efficiency.

*Table 7.* Ablation study of AOEPT under 70% image / both missing conditions.

| | Image Missing | | | Both Missing | | |
|---|---|---|---|---|---|---|
| | **MM-IMDb** | **HateMemes** | **Food101** | **MM-IMDb** | **HateMemes** | **Food101** |
| **Variant** | F1-M | AUC | ACC | F1-M | AUC | ACC |
| w/o MCP | 52.68 | 64.63 | 87.40 | 50.10 | 67.27 | 81.39 |
| w/o Instantiation | 53.37 | 64.68 | 87.53 | 51.41 | 64.58 | 81.93 |
| w/o Consistency | 53.78 | 66.05 | 87.53 | 51.93 | 68.15 | 82.98 |
| w/ Reconstruction | 35.52 | 65.49 | 80.68 | 36.79 | 66.87 | 77.64 |
| **AOEPT** | **54.86** | **67.96** | **88.86** | **53.11** | **69.80** | **83.24** |

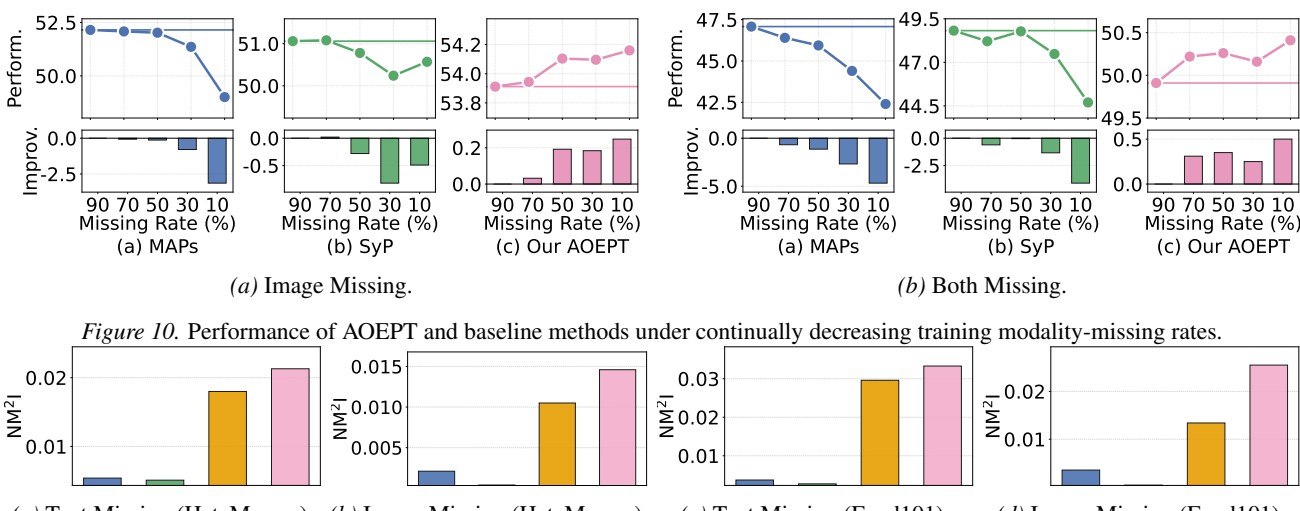

*(a)* Image Missing.  *(b)* Both Missing.

*Figure 10.* Performance of AOEPT and baseline methods under continually decreasing training modality-missing rates.

*(a)* Text Missing (HateMemes).  *(b)* Image Missing (HateMemes).  *(c)* Text Missing (Food101).  *(d)* Image Missing (Food101).

*Figure 11.* NM$^2$I comparison of AOEPT and baselines on the HateMemes and Food101 dataset under different missing-modality settings. Notably, the legend for this experiment is the same as the main paper, where the first two columns are baseline models MAPs and SyP, respectively, the third column is AOEPT w/o Instantiation, and the final column is AOEPT.

### G.2. Additional Ablation Studies

In this section, we provide the ablative results of AOEPT under image and both modality missing scenarios in Table 7. From the results, we draw the similar conclusion as in the main paper.

### G.3. Additional Modality Information Scaling Evaluation

We provide the evaluation of the modality information scaling under the image and both missing conditions in Figure 10. And we observe that, AOEPT can benefit from the additional available training information from the missing modality (decreasing training missing rate), while the performance of baselines plateau.

### G.4. Additional NM$^2$I Evaluation

We additionally provide the results of NM$^2$I evaluation on the HateMemes and Food101 datasets in Figure 11. And we observe that, AOEPT achieves the clearly higher NM$^2$I values across these two datasets comparing to the baselines.

### G.5. Performance of AOEPT on Tri-Modal Benchmark

To further evaluate the effectiveness of AOEPT in facing the modality-missing scenarios under multiple modalities setting, we conduct additional experiments on the tri-modal benchmark IEMOCAP using MulT (Tsai et al., 2019) as MT backbone. We then compare it with baseline MAPs. The IEMOCAP benchmark includes three modalities: audio (A), Video (V), and Text (T). Consequently, we design two set of modality-missing protocols: ① **Single-Modality Missing** at $\eta$%: $\eta$% of samples have exactly one modality missing (e.g., Audio indicates that the audio modality is missing), while the remaining samples are modality-complete. ② **Double-Modality Missing** at $\eta$%: $\eta$% of samples miss two modalities simultaneously

*Table 8.* Performance (%) under different modality-missing scenarios with a 70% missing rate on the tri-modal benchmark IEMOCAP. The best results are in **bold** and the second best are underlined.

| Method | Audio | | Video | | Text | | Audio-Video | | Audio-Text | | Video-Text | |
|---|---|---|---|---|---|---|---|---|---|---|---|---|
| | ACC | F1 | ACC | F1 | ACC | F1 | ACC | F1 | ACC | F1 | ACC | F1 |
| LB (**MulT**) | 53.41 | 53.40 | 57.46 | 54.63 | 56.50 | 54.28 | 54.90 | 54.05 | 45.95 | 41.65 | 55.33 | 52.12 |
| MAPs (Lee et al., 2023) | 54.16 | 53.64 | 59.81 | 57.89 | 57.89 | 55.35 | 50.85 | 50.45 | 46.27 | 42.32 | 55.86 | 52.44 |
| **AOEPT** | **55.12** | **54.57** | **61.73** | **59.60** | **58.64** | **56.81** | **56.18** | **55.69** | **48.08** | **44.66** | **59.70** | **55.63** |

(e.g., Audio–Video indicates that only the text modality is available), while the remaining samples are complete. As illustrated in Table 8, AOEPT outperforms all baselines across all missing settings.

## H. Relationship between AOEPT and Traditional Modality Missing Learning Methods

Traditional modality-missing learning methods mainly fall into two categories: Unified multimodal learning methods (Wang et al., 2023a; Zhao et al., 2021; Kim & Kim, 2024), and Modality imputation methods (Cai et al., 2018; Ma et al., 2021; Wang et al., 2023c;b). Despite their different technical implementations, these methods share a common objective: to preserve and exploit information from all modalities, even when some of them are absent at inference time. Unified multimodal learning methods aim to learn representations that are robust to modality absence by enforcing alignment or invariance across modalities. Modality imputation methods, on the other hand, explicitly recover the missing modality mainly through cross-modal generation, and then rely on the reconstructed signals (ground-truth representations of the missing modalities) for training. While effective, both paradigms often require tailored, specific architectural designs and additional networks.

In contrast, AOEPT does not explicitly enforce modality invariance nor perform explicit modality reconstruction. Instead, it internalizes the core principle underlying these traditional approaches: preserving access of MTs to information repositories of the missing modalities when prompting the MTs. By distilling global modality-wise contextual information into Modal-Contextualized Prompts and selectively instantiating them conditioned on the remaining modalities, AOEPT enables MTs to implicitly access and leverage information from missing modalities without altering the backbone architecture or introducing heavy reconstruction modules. From this perspective, AOEPT can be viewed as a *principled and lightweight instantiation of modality-missing learning under the MT framework*, which reformulates the core insights of traditional approaches into the unified and general multimodal model architecture (i.e., MTs).

## I. Literature Review for Prompt Learning

Prompt learning, a parameter-efficient fine-tuning strategy that adapts large-scale pretrained frozen backbone models (e.g., CLIP (Radford et al., 2021)) to downstream tasks by optimizing only a small set of learnable prompt parameters, has been widely adopted in the multimodal and computer vision communities (Zhou et al., 2022b;a; Liu et al., 2025; Wang et al., 2025). Pioneering study CoOP (Zhou et al., 2022b) introduced learnable prompt tokens into the language branch of CLIP, which are jointly optimized with image inputs to adapt the CLIP to downstream tasks, while CoCoOP (Zhou et al., 2022a) further leveraged the image inputs as conditions to derive the sample-specific prompts. Following studies such as ProGrad (Zhu et al., 2023) and KgCoOP (Yao et al., 2023) further explore how to align learnable prompts with the pretrained knowledge encoded in CLIP, aiming to preserve its generalization ability during prompt tuning. MaPLe (Khattak et al., 2023) extends prompt learning to both the visual and language branches of CLIP, enabling joint multimodal adaptation for improved downstream performance. DePT (Zhang et al., 2024) decouples the pretrained base knowledge from task-specific adaptations during prompt tuning, mitigating interference between general and downstream-oriented representations. SurPL (Liu et al., 2025) learned a single base prompt and employs a lightweight surrogate feature generator to produce diverse prompted text features from it, bypassing the issue of enormous gradient computation inside the text encoder. With the success of prompt learning in adapting vision–language models to downstream tasks, recent studies (Lee et al., 2023; Hu et al., 2024; Zhao et al., 2025; Lang et al., 2025a) have begun to adopt this parameter-efficient strategy to enhance the robustness of Multimodal Transformers (MTs) under modality-missing scenarios, where the incomplete inputs and learnable prompts are fed to the MTs to perform the prompt tuning. Building upon this line of research, we identify an inherent limitation in existing methods, namely Implicit Modality-Reduction, and propose AOEPT, a lightweight missing-adaptive modal-contextualized prompting framework that effectively mitigates this bottleneck.

