# OpenReview forum: "AOEPT: Breaking the Implicit Modality-Reduction Bottleneck in Modality-Missing Prompt Tuning"
_ICML.cc/2026/Conference — ICML 2026 regular_

### Official Review · Reviewer_FZVz · 2026-02-18

**Soundness:** 3
**Presentation:** 3
**Significance:** 3
**Originality:** 3
**Overall Recommendation:** 4
**Confidence:** 4

**Summary:**

This paper identifies an "Implicit Modality-Reduction (IMR) bottleneck" in existing prompt-tuning methods for multimodal
  transformers (MTs) under missing-modality scenarios: because existing methods generate prompts solely from observed modality
  signals, the prompts are statistically independent of the missing modality's information by construction, limiting the MT's
  reasoning to the reduced observed-modality subspace. To address this, the authors propose AOEPT, which distills global
  modality-specific contextual information from training data into Modal-Contextualized Prompts (MCPs). At inference time, MCPs are
  instantiated per sample by conditioning on the remaining observed modalities via a learned gating mechanism, selectively activating
   relevant missing-modality information. An intra-modal latent consistency regularizer encourages the instantiated prompts to align
  with the missing modality's actual latent representations. The method is evaluated on MM-IMDb, HateMemes, Food101 (dual-modal) and
  IEMOCAP (tri-modal) across CLIP, ViLT, and MulT backbones, with a proposed NM2I diagnostic metric to empirically verify bottleneck
  alleviation.

**Compliance With Llm Reviewing Policy:**

Affirmed.

**Final Justification:**

The author has answered most of my questions, and I will raise my rating.

**Key Questions For Authors:**

---
  Key Questions For Authors

  1. Non-degeneracy condition (critical): Proposition F.2 assumes G_ψ does not ignore C_t given z. Is there a training condition
  (e.g., on the consistency loss weight or the learning dynamics) that provably prevents G_ψ from collapsing to ignoring C_t? If a
  response can provide such a condition or an argument for why collapse does not occur in practice, it would significantly strengthen
   the theoretical contribution.
  2. NM2I as a performance proxy: Is there a consistent monotonic relationship between NM2I and downstream task accuracy across all
  conditions (different missing rates, backbones, datasets)? Could the authors provide a scatter plot of NM2I vs. accuracy across
  experimental conditions? If NM2I does not consistently predict performance, its value as a diagnostic metric should be qualified.

**Limitations:**

The authors mention limitations briefly in one paragraph and note that "more advanced MCP designs and instance-aware instantiation
  mechanisms warrant further investigation." This is insufficient. The paper should discuss: (1) failure modes of the NM2I metric as
  a performance proxy; (2) settings where the IMR bottleneck may not be the dominant failure mode (e.g., low missing rates); (3) the
  assumption that training-set modality distributions are representative of test-time distributions. Constructive suggestion: add a
  dedicated limitations subsection with 2–3 concrete failure-mode analyses.

**Strengths And Weaknesses:**

Strengths:

  Soundness. The core claim — that existing prompt methods cannot carry instance-level missing-modality information by construction —
   is rigorously supported by Lemma F.1 and validated empirically through two complementary tools: the pilot experiment (Figure 4,
  showing that simply initializing MAPs prompts with modality priors improves performance) and the NM2I metric (Figure 8, showing
  near-zero information sharing in baselines). The ablation study (Table 3) systematically isolates the contribution of each
  component (MCPs, instantiation, consistency regularization), and results across 50%/70%/90% missing rates (Tables 2 and 4)
  demonstrate robustness to varying conditions.

  Presentation. The paper is clearly written and well-structured. The problem motivation flows naturally from the theoretical
  observation to the proposed solution. Figures 2–3 effectively illustrate the AOEPT workflow.

  ---
  Weaknesses:

  Soundness. Proposition F.2 proves that AOEPT establishes an information-access path "under a mild non-degeneracy condition that G_ψ
   does not ignore C_t given z." This condition is assumed rather than proved, with NM2I > 0 serving as post-hoc empirical
  verification. The theoretical justification is thus partly circular: we assume the method works (non-degeneracy), then show
  empirically that it does (NM2I > 0). A formal sufficient condition on the training objective guaranteeing non-degeneracy would
  strengthen the theory.

  Significance. Improvements over SyP — the strongest baseline — are often modest at the 70% missing rate setting: e.g., 51.50 vs.
  49.68 F1-M on MM-IMDb text-missing (CLIP), 71.12 vs. 68.94 AUROC on HateMemes text-missing. The advantage is more pronounced in the
   modality scaling analysis but is less compelling in head-to-head comparison at standard settings.

---

> ### Author Rebuttal · Authors · 2026-03-31
>
> > Q1: Non-degeneracy condition (critical): Proposition F.2
>
> Thank you for your thoughtful question. In Proposition F.2, we formulate the prompt construction in AOEPT as:
> $$
> \mathbf{P} = \mathcal{G}_\psi(z, \mathbf{C}_t),
> $$
>
> where the non-degeneracy condition assumes that $\mathcal{G}\_\psi$ does not ignore $\mathbf{C}\_t$ and collapse to $\mathbf{P} = \mathcal{G}\_\psi(z)$. In our design, such collapse is prevented both by the design of $\mathcal{G}\_\psi$ and by the optimization constraints.
>
> Specifically, $\mathcal{G}_\psi$ is implemented as a **conditional gated network**, where the modality-level repository $\mathbf{C}_t$ serves as the **primary information source**, and the observed modality $z$ only **modulates and decodes** relevant information from $\mathbf{C}_t$ (cf. Eq.(8)-(9) for details). Thus, $\mathbf{C}_t$ cannot be bypassed through the prompt construction.
>
> Furthermore, the proposed **intra-modal latent consistency regularization** (cf. Eq.(9)) explicitly aligns the instantiated prompts with ground-truth missing-modality representations, encouraging $\mathcal{G}_\psi$ to effectively utilize information from $\mathbf{C}_t$ and preventing degeneration.
>
> In addition, both the ablation study (variant w/o MCP in Table 3) and the NM2I analysis (Fig. 10) provide empirical evidence that further supports this claim.
>
> > Q2: The role of NM2I metric and its application.
>
> Thank you for your insightful question.
>
> We would like to clarify that, as mentioned in Sec 4.7 NMI Analysis, **NM2I is not designed as a performance proxy**, and we do not assume a strictly monotonic relationship between NM2I and downstream accuracy across all settings. Instead, NM2I is introduced as a **diagnostic metric for the IMR bottleneck**, measuring whether the learned prompts encode information related to the missing modality. In other words, it only assesses whether a method suffers from or alleviates the IMR bottleneck, rather than to directly predict task performance.
>
> In practice, we observe that higher NM2I (e.g., AOEPT, and MAPs in Sec. 4.2 Pilot Experiment) is associated with a higher potential for better performance, supporting our claim that **enabling explicit access to missing-modality information space is beneficial**. However, downstream performance (accuracy) is also influenced by other factors, such as deeper exploitation of unimodal information through delicate prompts (e.g., correlated prompting in DCP and synergistic prompting in SyP), so strict monotonicity is neither expected nor required. Below, we provide the NM2I results for all baselines on MM-IMDb dataset with text/image missing.
>
> |MM-IMDb(NM2I)|Text|Image|
> |-|-|-|
> |MAPs|1.8e-3|1.5e-3|
> |DCP|9.3e-4|6.7e-4|
> |RAGPT|3.0e-3|2.4e-3|
> |MemPrompt|2.6e-3|2.3e-3|
> |SyP|1.5e-4|1.8e-4|
> |PROMISE|1.0e-4|1.0e-4|
> |AOEPT|1.6e-2|1.4e-2|
>
> We sincerely thank you for the suggestion and will further clarify it to avoid any ambiguity in the revised version.
>
> > Q3: Add a dedicated limitations subsection with 2–3 concrete failure-mode analyses.
>
> We sincerely thank you for your suggestions and will include a dedicated limitation section in the revised version, covering all the aspects discussed in prior response.

---

> > ### Author Rebuttal · Reviewer_FZVz · 2026-04-02
> >
> > I appreciate the authors' detailed clarifications and their acknowledgment of the weaknesses in the initial presentation. I am generally satisfied with the proposed changes and am willing to increase my presentation score once these changes are reflected in the revised manuscript.

---

> > > ### Author Response · Authors · 2026-04-03
> > >
> > > Thank you for your feedback and your time. We are glad that most concerns are resolved.

---

### Official Review · Reviewer_pwEe · 2026-03-03

**Soundness:** 2
**Presentation:** 2
**Significance:** 2
**Originality:** 2
**Overall Recommendation:** 3
**Confidence:** 4

**Summary:**

This paper addresses the challenge of handling missing modalities in real-world multimodal systems, with a specific focus on Multimodal Transformer (MT) architectures. The authors identify a critical flaw in existing prompt-tuning methods, which they term the "Implicit Modality-Reduction bottleneck." Specifically, current approaches condition prompts solely on the available modalities, inadvertently restricting the model's reasoning capacity to a reduced subspace and cutting off access to latent information from the missing modalities.
To overcome this limitation, the authors propose AOEPT, a novel framework based on modal-contextualized prompting. The core innovation lies in the introduction of Modal-Contextualized Prompts (MCPs), which are designed to distill and store global modality-specific priors from the training data, acting as latent repositories for the missing information. During inference, these MCPs are conditioned on the remaining observed modalities to generate instance-aware prompts. This mechanism effectively augments the missing-modality information for individual samples, restoring the MT's reasoning scope beyond the observed-only subspace.

**Compliance With Llm Reviewing Policy:**

Affirmed.

**Final Justification:**

Thank you for the detailed rebuttal and the additional analysis. The rebuttal clarifies that IMR is intended as a limitation rather than a failure mode, but this also weakens the claimed significance of identifying IMR as the paper’s core contribution. The response explains why large gains may be difficult to obtain, but it still does not resolve the central tension: if IMR is a major bottleneck, then the empirical gains are too modest; if the gains are inherently modest, then the bottleneck appears less significant than the paper suggests.

In particular, the upper-bound argument is not sufficient to justify the importance of IMR. Showing that there is limited headroom does not establish that IMR is a truly important bottleneck whose resolution should be viewed as a central contribution. If IMR were indeed as important as claimed, I would expect stronger empirical evidence, potentially including datasets or benchmarks where addressing it leads to more clearly substantive improvements.

Relatedly, the contradiction between the near-zero NM2I of baselines and their competitive final performance remains insufficiently explained. The clarification regarding MemPrompt is helpful. Overall, my main concern about the soundness and significance of the paper’s core contribution remains unresolved.

**Key Questions For Authors:**

Figure 1 can be improved; the information in Figure 1 is unclear: why did the color of the mid-stream of the right figure change?


In section 4.2, pilot experiment, "We observe performance improvements with these modified prompts, indicating that the original performance of MTs is bound to the degraded, single modality input structure, despite their strong pretrained multimodal modeling capacity." The setting is unclear. Are the results obtained after training or not?

**Limitations:**

yes

**Strengths And Weaknesses:**

Pros:

The setting and motivation are clear.

The authors have conducted several experiments on several benchmarks to show the effectiveness of their methods.


Cons:

What is the relation between this work and MemPrompt, which leverages a prompt memory bank to restore the missing modality prompts? It looks like sharing a similar idea to this work.

Do all the baselines suffer from the "Implicit Modality-Reduction Bottleneck" or just the MAP?

- If only MAP suffers from this limitation, considering that MAP dates back to 2023, and other works do not suffer from this limitation, makes this finding less significant.

- If all previous methods suffer from this limitation, which is great, it means that all previsou method overlook the initialization issue. However, in this case, the previous method achieves very close performance compared with the proposed method, which violates the assumption of this limitation.

---

> ### Author Rebuttal · Authors · 2026-03-31
>
> > W1: The relation between this work and MemPrompt.
>
> Thank you for your question. We would like to clarify that, although MemPrompt incorporates memory components, it still falls within the Implicit Modality-Reduction (IMR) bottleneck, as the memory prompts are randomly initialized and do not encode missing-modality information.
>
> In contrast, AOEPT explicitly models modality-level repositories via **Modal-Contextualized Prompts (MCPs)** and performs **instance-aware instantiation**, thereby alleviating the IMR bottleneck by restoring the reasoning scope of the Multimodal Transformer (MT) to the full modality space. For a more detailed explanation, we kindly refer you to the response to the Reviewer H1hy Q1.
>
> To further distinguish MemPrompt from the proposed AOEPT, **below we conduct additional experiments measuring the NM2I of MemPrompt** and compare it with AOEPT (reported in Sec. 4.7 NMI Analysis for Implicit Modality-Reduction). Additional NM2I results for other baselines are provided in response to Reviewer FZVz Q2. The results demonstrate that AOEPT achieves substantially higher NM2I, confirming that AOEPT alleviates the IMR bottleneck whereas MemPrompt does not.
>
> |MM-IMDb(NM2I)|Text|Image|
> |-|-|-|
> |MemPrompt|2.6e-3|2.3e-3|
> |AOEPT|1.6e-2|1.4e-2|
>
> > W2: All baselines suffer from the "Implicit Modality-Reduction Bottleneck" or just the MAP, and the performance gains.
>
> Thank you for your question. Firstly, as mentioned in the paper **Sec. 1 (Introduction) and Sec. 2 (Related Work: Modality-Missing Learning)**, the IMR bottleneck applies to all **mainstream prompt-tuning methods**, including MAP and related variant baselines, where prompts are either randomly initialized or conditioned only on observed modalities. We kindly refer you to the discussion of baselines about the IMR bottleneck in the response to the Reviewer mRBt Q1.
>
> Moreover, IMR is a **limitation on information accessibility**, rather than a complete failure mode. Existing methods can still achieve reasonable performance by exploiting signals within the reduced modality space. Nevertheless, AOEPT delivers **consistent improvements across 3 architectures (CLIP, ViLT, MulT), 4 datasets (MM-IMDb, HateMemes, Food101, IEMOCAP), and several missing settings**, demonstrating that restoring access to missing-modality information has the potential in enhancing performance under a lightweight design. Importantly, our contribution goes beyond empirical gains: we **identify and address a previously under-explored bottleneck** in prompt-based modality-missing learning for MTs.
>
> > Q1: The color of the mid-stream of the right figure change.
>
> Thank you for your question, and we sincerely apologize for any confusion caused by the right figure in Fig. 1.
>
> In prompt-tuning, the **learnable prompt tokens** will interact with the **observed modality tokens** within the transformer layers [1]. Initially, the missing modality (e.g., text in the figure) is shown in gray, as it is **padded with dummy values**. In AOEPT, the prompts are explicitly initialized from the missing-modality space and thus are represented using the corresponding modality color.
> Accordingly, after the interaction, we **update the color of the dummy modality tokens** to match that of the prompt tokens, indicating that they now carry information of the missing modality.
>
> We will clarify this design choice more explicitly in the figure caption in the revised version.
>
> > Q2: The setting in the pilot experiment: whether the results are obtained after or before training.
>
> Thank you for your question, and we sincerely apologize for any confusion. As clarified in the Sec. 1 (Introduction) and Sec. 4.2 (Pilot Experiment), we replace the randomly initialized learnable prompts with clustered modality representations for initialization. Therefore, the standard MAPs training procedure is followed, and the results are **obtained after training**.
>
> And this pilot experiment aims to compare two initialization strategies for the prompts: random initialization versus missing modality prior. The results demonstrate that incorporating missing modality priors into prompt initialization yields a further performance gain over random initialization, underscoring the importance of introducing such priors into the prompt tuning framework.
>
> We apologize again and will further refine the description to avoid any ambiguity.
>
> ## Reference
>
> [1] Visual prompt tuning, ECCV'22.

---

> > ### Author Rebuttal · Reviewer_pwEe · 2026-04-04
> >
> > I appreciate the authors' detailed response and the additional experiments.
> >
> > My W2 raised a deliberate dilemma, and I note that the authors' response inadvertently confirms my concern rather than resolving it. Specifically:
> >
> > - The authors claim that all existing methods suffer from the IMR bottleneck — this is the stronger and more interesting claim. However, they then qualify IMR as "a limitation on information accessibility, rather than a complete failure mode." This softening is necessary to explain why existing methods already achieve competitive performance, but it simultaneously undermines the significance of IMR as the paper's core motivation and contribution. If the bottleneck is not a critical failure mode, then framing the entire paper around "identifying and addressing" it appears to be an overclaim.
> >
> > - The paper's narrative suggests that IMR is a fundamental, previously overlooked problem that severely limits existing methods. Yet the empirical evidence tells a different story — the performance gaps between AOEPT and baselines are often marginal. The authors cannot have it both ways: IMR is either a significant bottleneck (in which case, where are the substantial gains?) or it is a mild limitation (in which case, the contribution is incremental at best).
> >
> > - The authors emphasize "consistent improvements across 3 architectures, 4 datasets," but consistency of small improvements does not establish the importance of the identified bottleneck. It could equally suggest that the IMR bottleneck is a minor factor among many, and the proposed solution provides only marginal additional signal.
> >
> > - "baseline methods yield nearly zero NM2I" which is contradictory to the close performance between the proposed method and other baselines. This implies that IMR is not important to final performance, or the proposed solution does not address the IMR properly.
> >
> > On W1 (MemPrompt): I acknowledge the NM2I comparison, which helps distinguish the two methods quantitatively. However, the conceptual overlap (using a memory mechanism to compensate for missing modality information in prompts) still makes the novelty less clear-cut than the authors suggest.
> >
> > Summary: My primary concern is not about individual experimental results, but about the soundness of the paper's core contribution. The IMR bottleneck is presented as a significant, previously unrecognized problem, yet the empirical evidence fails to convincingly support this level of significance. In its current form, I do not find the work meets the acceptance threshold.
> >
> > I keep my scores unchanged.

---

> > > ### Author Response · Authors · 2026-04-06
> > >
> > > We sincerely thank you for the thoughtful follow-up comments. Below, we carefully clarify from: (1) significance of IMR, (2) performance gains, and (3) distinction from MemPrompt.
> > >
> > > 1. **Significance of IMR**.
> > >
> > > 	In the paper, we do not claim that IMR is a complete failure mode, but rather a **limitation on reduced modality information accessibility**. In multimodal tasks (including modality-missing settings), models are expected to reason over the full modality space (e.g., reconstruction-based methods [1] aim to recover the missing modalities to restore this capability), whereas most prompt-tuning methods implicitly restrict MTs to the observed modality subspace.
> > >
> > > 	Importantly, this does not imply performance collapse. Due to modality relationships such as modality redundancy [2], MTs can still achieve reasonable performance from reduced-modality space. This is the reason we refer to it as a bottleneck: it hinders the utilization of missing-modality information space, rather than introducing a catastrophic failure mode.
> > >
> > > 2. **Performance gains**.
> > >
> > > 	Although the notion of "substantial" improvements can be subjective, we provide two relatively objective views.
> > >
> > > 	+ Firstly, below we provide the upper bound (UB) performance of MT (CLIP) for modality-missing (i.e., MT is prompt-tuned and evaluated on modality-complete datasets), which serves as a commonly used and natural upper bound. We observe that both AOEPT and the strongest baselines are already close to this upper bound, leaving **relatively limited room** for drastic gains (T: Text, I: Image, B: Both).
> > >
> > > 	||MM-IMDb (T)|I|B|HateMemes (T)|I|B|Food101 (T)|I|B|
> > > 	|-|-|-|-|-|-|-|-|-|-|
> > > 	|DCP (NeurIPS'24)|49.99|52.77|50.70|62.82|64.12|66.08|78.87|87.32|81.87|
> > > 	|SyP (ICCV'25)|49.68|53.19|52.77|68.94|66.98|68.42|79.56|88.67|82.45|
> > > 	|AOEPT|51.50|54.86|53.31|71.12|67.96|69.80|80.77|88.86|83.24|
> > > 	|UB|58.00|58.00|58.00|72.02|72.02|72.02|91.88|91.88|91.88|
> > >
> > > 	+ In addition, we analyze the relative improvements on each dataset. On MM-IMDb and Food101, the year-over-year SOTA progression from **DCP → SyP** is **1.42%** and **1.04%**, respectively, while the gain from **SyP → AOEPT** is **2.61%** and **0.90%**, demonstrating that our improvements are **comparable to or even exceed those between recent SOTA methods**, despite the **smaller remaining room for improvement**. For HateMemes, although the relative gain from DCP to SyP is larger, both **SyP and AOEPT already achieve performance very close** to the upper bound, whereas DCP remains significantly below it.
> > >
> > > 	Furthermore, most existing methods struggle to benefit from improved training conditions (i.e., lower missing rates), as they remain confined to the observed modality subspace (IMR bottleneck). In contrast, AOEPT alleviates IMR and exhibits clear further improvements (cf. Sec. 4.8 Modality Information Scaling) by distilling richer modality-level distributions from the training data.
> > >
> > > 3. **Distinction from MemPrompt**.
> > > 	We thank you for raising this point again and apologize for any confusion. MemPrompt and AOEPT are fundamentally different, and that MemPrompt does not explicitly compensate for the reduced missing-modality space for alleviating the IMR bottleneck. Below, we list the core differences:
> > > 	+ **Prompt Construction**: MemPrompt builds a memory bank of multiple randomly initialized prompts, whereas AOEPT models MCPs (Modal-Contextualized Prompts) derived from training data as global modality proxies. Moreover, AOEPT designs a **single prompt structure** without relying on **prompt-level memory**, resulting in a more compact and lightweight design.
> > > 	+ **Prompt Utilization & Optimization**: MemPrompt generates final prompts via similarity-based retrieval and aggregation over randomly initialized memory prompts, and is optimized solely with the task objective, without any explicit supervision for compensating information from the reduced modality space (reflected in the NM2I results). In contrast, AOEPT performs instance-aware instantiation of MCPs conditioned on observed modalities, enabling controlled access to the missing-modality space, and further supervised by intra-modal latent consistency regularization for improved fidelity.
> > > 	+ **Efficiency & Performance**: MemPrompt requires a prompt memory and performs retrieval and aggregation, incurring additional overhead (cf. Sec. 4.9 Efficiency Analysis), while AOEPT achieves better performance with a more efficient design.
> > >
> > > 	Overall, AOEPT is not a variant of memory-based prompt tuning method, but a lightweight method that enables controlled accessibility of each sample to its reduced modality space.
> > >
> > > We sincerely thank you again for your valuable feedback, and we hope that these clarifications are helpful in alleviating your remaining concerns.
> > >
> > > # Reference
> > > [1] RAG4DMC: Retrieval-Augmented Generation for Data-Level Modality Completion, ICLR'26
> > >
> > > [2] I2MoE: Interpretable Multimodal Interaction-aware Mixture-of-Experts, ICML'25

---

### Official Review · Reviewer_mRBt · 2026-03-05

**Soundness:** 4
**Presentation:** 4
**Significance:** 3
**Originality:** 2
**Overall Recommendation:** 4
**Confidence:** 3

**Summary:**

This paper addresses the problem of robust adaptation for Multimodal Transformers under conditions of missing modalities during both training and inference. The authors argue that conventional Prompt Tuning methods tend to fall into the local latent space of the remaining modalities, leading to a "modality-reduction" bottleneck. To tackle this, they propose the AOEPT framework. Its core contributions include the construction of a Modality-prior Centroid Pool (MCP) via clustering refinement, combined with sample-adaptive prompt instantiation and latent consistency regularization. The method aims to compensate for missing semantic information using external knowledge. Experiments conducted on various backbones (e.g., CLIP, ViLT) across different missing rates demonstrate the effectiveness, performance superiority, and representational stability of the proposed approach.

**Compliance With Llm Reviewing Policy:**

Affirmed.

**Final Justification:**

The author has answered most of my questions, and I will raise my rating.

**Key Questions For Authors:**

* **Q1. Precisely define the scope of your "bottleneck" and clarify boundaries with prior work:** Which specific baselines are covered by your "observed-only prompting" assumption? For prompt methods involving memory, retrieval, or compensation mechanisms (where information does not come solely from the observed modality), do you believe they still suffer from the same bottleneck? If yes, please provide a clearer justification; if not, please refine the scope of your claim to avoid over-generalizing that "all existing methods are forced into unimodal reasoning." A more systematic taxonomy would better support your novelty claim.
* **Q2. Clarify the meaning of "escaping the subspace": Prior Accessibility vs. Instance-level Recovery?** The analysis suggests you do not guarantee exact recovery of missing data. Please clarify: Is the "reasoning scope expansion" intended to mean *accessibility to relevant priors* or *instance-level latent compensation*? If the former, please moderate the phrasing in the text that implies "instance-level recovery" and explain why prior injection alone is sufficient to be called "escaping the subspace."
* **Q3. Provide a diagnostic to prove the gate does not degenerate into a "constant gate + global average prior":** To counter the concern that the observed modality dominates the output, please provide statistics for the gate (e.g., variance/entropy across samples, changes across different missing rates). Does the gate maintain sufficient sample discriminability under high missing rates?
* **Q4. Add a direct comparison with "Memory/Retrieval-based Prior Injection":** Given that AOEPT can be viewed as an offline modality prior pool with conditional activation, how does it differ fundamentally from representative memory/retrieval prompt methods? Please explain why your MCP (layer-wise, k-means prototypes) offers a unique mechanistic advantage rather than just an engineering variation.

**Overall Assessment:** This paper is nearly "impeccable" in all aspects except for the depth of its insight and the rigor of its novelty positioning. I will consider raising my score if the authors can convincingly address the questions above.

**Limitations:**

Yes

**Strengths And Weaknesses:**

### Strengths

* **S1. Clear Problem Definition and Practical Significance:** The paper focuses on scenarios where modalities may be missing during both training and inference (covering Text Missing, Image Missing, and Both Missing). This is a common yet challenging issue in real-world systems that lacks a unified, efficient solution. The narrative flows logically from "missing-induced inference degradation" to a concrete solution.
* **S2. Concise, Modular Design with Low Computational Overhead:** AOEPT injects missing modality priors into the prompt tuning framework via the "MCP + Instantiation Gating" mechanism. It does not rely on heavy reconstruction networks or large generative models, maintaining the lightweight nature of prompt-tuning during inference. The offline pool construction and online instantiation make it easy to deploy and integrate with existing Multimodal Transformer backbones.
* **S3. Comprehensive Experiments and Stable Gains:** The authors evaluate the method across multiple datasets, two types of backbones (CLIP/ViLT), and various missing patterns/rates. The results show consistent advantages even at high missing rates, proving the robustness of the method. The ablation studies effectively validate the contribution of each component (MCP, instantiation, and consistency regularization).
* **S4. Clear Writing and Sufficient Implementation Details:** The paper provides clear descriptions of modality processing, training/inference pipelines, and complexity analysis. The organization is standard and logical, making it easy for reviewers to verify the relationship between contributions and conclusions.

### Weaknesses

* **W1. Over-generalization Risk of the Key Insight (Implicit Modality-Reduction Bottleneck):** The paper categorizes existing prompt tuning methods into two types: randomly initialized prompts or sample-specific prompts conditioned on observed modalities. It asserts that these mechanisms inevitably "compress" multimodal reasoning into a unimodal subspace. This is a strong claim that may invite scrutiny regarding: (i) whether it truly covers all related work, and (ii) whether the relationship with existing memory/retrieval/compensation methods is oversimplified. Specifically, while the paper mentions works that utilize "memory/retrieval" to compensate for missing info, it does not clearly define whether these also fall into the "observed-only prompting" bottleneck, potentially blurring the boundaries of its novelty and positioning.
* **W2. Potential Tension Between the "Escape Subspace" Narrative and Actual Mechanism:** AOEPT still instantiates prompts based on the observed modality representation (via the gating mechanism). While the information-theoretic analysis highlights the advantage of "external prior paths," it also notes that this does not guarantee the recovery of exact instance-level missing data. Thus, claims such as "expanding reasoning scope" or "fully triggering multimodal capacity" might be perceived as stronger than what is actually achieved. There is a lack of direct empirical diagnosis to show whether the gate might degenerate into an almost constant value when the correlation between modalities is low, effectively reverting the method to a "unimodal + global prior" model rather than true cross-modal reasoning.

---

> ### Author Rebuttal · Authors · 2026-03-31
>
> > Q1: The scope of the Implicit Modality-Reduction (IMR) bottleneck and clarify boundaries with prior work.
>
> We appreciate this question and take this opportunity to further clarify the boundaries with prior work.
> Below, we divide the group of studies for modality-missing in Multimodal Transformer (MT) into three subgroups:
>
> 1. As shown in Sec. 2 (Related Work) and Sec. 3.1 (Preliminary), **mainstream prompt-tuning baselines** all fall within the scope of the IMR bottleneck, as their prompts are either randomly initialized or conditioned solely on the observed modalities.
> 2. For the **memory-based baseline** MemPrompt, although it has memory components, the memory prompts are still randomly initialized. Thus, it also falls within the IMR bottleneck.
> 3. For the **retrieval-augmented baseline** RAGPT, as shown in Sec. 2 (Related Work) and Sec. 4.3 (Main Performance), it incorporates retrieved multimodal information mainly for **missing modality reconstruction**, therefore **differs** from standard prompt-tuning methods, which are typically designed to be lightweight (Evidenced by Sec. 4.9. Efficiency Analysis). Moreover, it does not explicitly model the IMR bottleneck, resulting in a heavy yet less effective design. It relies on costly instance-level retrieval with off-the-shelf encoders, lacking task-specific optimization. In addition, such **hard external retrieval** is inherently bounded to a limited subset of samples, leading to incomplete and potentially biased information for the reduced modality space.
>
> Moreover, we kindly refer you to the NM2I results for these models in response to Reviewer FZVz Q2 for further empirical evidence.
>
> We will further clarify the boundaries in the revised paper.
>
> > Q2: The meaning of escaping the subspace and prior accessibility vs. instance-level recovery.
>
> Thank you for the question. We clarify that AOEPT enables both modality-level prior accessibility and instance-level reasoning scope compensation.
>
> 1. First, it **establishes modality-level prior accessibility** via Modal-Contextualized Prompts (MCPs) (Eq.(2)-(5)), which make missing-modality information globally available to the MT. At this stage, the reasoning scope of the MT is **expanded toward the full modality space**, alleviating the IMR bottleneck. Due to space limitations, we kindly refer you to response to Reviewer H1hy Q1.
>
> 2. Second, based on this accessibility and considering the characteristics of different samples, it performs **instance-aware instantiation** conditioned on the remaining modalities and constrained by the **intra-modal latent consistency regularization** (Eq.(8)-(9)), enabling adaptive and sample-specific utilization of the most relevant modality information in a lightweight manner.
>
> > Q3: A diagnostic to prove the gate does not degenerate into a constant gate.
>
> Thank you for the question. The variant w/o Instantiation in both Ablation study and NM2I analysis shows the effectiveness of the gating mechanism in enabling instance-level compensation.
>
> To further address your concerns, we provide gate-level diagnostics on **MM-IMDb under the text-missing scenario**, focusing on the vision conditioned gate that modulates the global Text-Contextualized Prompts.
>
> We calculate (1) the variance of per-channel gate values across samples and report the channel-averaged results, and (2) the averaged $L_2$ distance between the per-sample gate vector and the averaged gate vector across samples.
>
> |Missing Rate|Var|Distance|
> |-|-|-|
> |70%|3.20e-2|3.70|
> |90%|3.60e-2|4.08|
>
> Since the gate outputs are sigmoid-normalized, these values are clearly non-trivial and **far from those of a constant gate** (near zero Var and Dist), indicating non-degenerate behavior. Moreover, both variance and distance increase with higher missing rates as more samples require modulation, further confirming effective instance-specific gating.
>
> > Q4: A direct comparison with "Memory/Retrieval-based Prior Injection"
>
> Thank you for the question. We have provided a detailed discussion of memory- and retrieval-based prior injection in Response to Q1. In brief, memory-based methods (MemPrompt) still fall within the IMR bottleneck, while retrieval-based methods (RAGPT) incur substantial computational overhead and suffer from suboptimal performance due to noisy external retrieval.
>
> In contrast, AOEPT directly targets the IMR bottleneck by maintaining a latent, modality-level repository that encodes global modality information into a condensed representation space, enabling implicit, internalized retrieval that is inherently more robust and noise-tolerant.
>
> This distinction is also conceptually aligned with recent advances in parameter-efficient in-context test-time learning [1], where information is internalized into model parameters rather than accessed via external retrieval.
>
> We will further refine this comparison in the revised paper.
>
> ## Reference
>
> [1] PERK: Long-Context Reasoning as Parameter-Efficient Test-Time Learning, ICLR'26.

---

> > ### Author Rebuttal · Reviewer_mRBt · 2026-04-02
> >
> > The author has answered most of my questions, and I will raise my rating.

---

> > > ### Author Response · Authors · 2026-04-03
> > >
> > > Thank you for your positive feedback to our work and for raising your score. We appreciate your time and consideration.

---

### Official Review · Reviewer_H1hy · 2026-03-12

**Soundness:** 3
**Presentation:** 3
**Significance:** 3
**Originality:** 3
**Overall Recommendation:** 4
**Confidence:** 3

**Summary:**

This paper studies prompt tuning for multimodal transformers under missing-modality settings. The authors argue that existing methods rely only on observed modalities when constructing prompts, which leads to an implicit modality-reduction bottleneck. To address this issue, they propose AOEPT, which builds Modal-Contextualized Prompts (MCPs) from global modality information in the training set and then instantiates sample-specific prompts conditioned on the available modalities. Experiments on several benchmarks show that AOEPT consistently outperforms prior prompt-based methods across different missing-modality scenarios.

**Compliance With Llm Reviewing Policy:**

Affirmed.

**Final Justification:**

After reading other reviewers' comments and the response from the authors, I would like to maintain the scroring of weak accept, and I think this paper deserves the acceptance.

**Key Questions For Authors:**

1. How can you more directly verify that the performance gains come from mitigating the modality-reduction bottleneck, rather than from stronger priors or increased prompt flexibility?

2. How robust is AOEPT when the test distribution differs from the training distribution, especially for the missing modalities?

3. Have you compared against stronger baselines augmented with similar global modality priors or memory components?

**Limitations:**

yes.

**Strengths And Weaknesses:**

The paper is motivated by a clear and interesting insight: current prompt tuning methods for missing modalities may overly restrict the model to observed-modal information. The proposed method is conceptually simple and well aligned with this motivation. Empirically, the paper is strong: it evaluates on multiple datasets and backbones, includes extensive ablations, and provides additional analyses on robustness and efficiency. Overall, the method appears effective and practically lightweight.

The main conceptual claim—the so-called implicit modality-reduction bottleneck—is plausible, but the current evidence is still more suggestive than definitive. Some of the gains may also come from stronger global priors or a more favorable prompt parameterization, rather than uniquely from the proposed mechanism. In addition, the method depends on training-set-level modality repositories, so its robustness under distribution shift or low-resource settings is not yet fully examined. Finally, the paper’s claim of scalability to more modalities is only partially supported, since the tri-modal evidence is relatively limited.

---

> ### Author Rebuttal · Authors · 2026-03-31
>
> We thank you for valuable questions, and address them below:
>
> > Q1: The gains from mitigating IMR, or just from stronger priors and more flexible prompts?
>
> We apologize for any ambiguity in our paper that may have led to misunderstanding regarding IMR and our proposed method. The performance gains stem from **mitigating the IMR bottleneck**, which is evidenced by the Sec. 4.4 (Ablation Study, variant w/o MCP), the Sec. 4.7 (NMI Analysis) and the Sec. 4.8 (Modality Information Scaling). Specifically, AOEPT enables **controlled accessibility of each sample to its reduced modality space**, thereby mitigating the IMR bottleneck.
>
> Following, we answer this question from: (1) **Definition of Implicit Modality-Reduction (IMR) bottleneck**, (2) **Solution AOEPT**, and (3) **Empirical result**.
>
> 1. **IMR Bottleneck**. The IMR bottleneck is fundamentally a **modality-level** constraint in existing prompt-tuning for modality-missing methods: although the Multimodal Transformer (MT) is designed to operate over the *complete modality space*, most existing methods implicitly restrict it to a **reduced modality subspace** when modalities are missing.
>
> 	Specifically, consider a dual-modal setting with modalities $\mathcal{X}_1$ and $\mathcal{X}_2$, where each instance is $X = (x_1, x_2)$. When modality missing, e.g., $X = (x_1, *),$ most existing methods construct prompts $P$ by either random initialization or conditioning on the remaining modalities.
> 	$$
> 	P \sim \mathcal{D}~(\text{Random}) \quad \text{or} \quad P = C(x_1),
> 	$$
>
> 	The MT then performs prediction with only $x_1$ and $P$. As a result, the inference space of the MT collapses from the $\mathcal{X}_1 \times \mathcal{X}_2$ to a reduced subspace $\mathcal{X}_1,$ indicating that the MT **cannot access or reason over information associated with modality $\mathcal{X}_2$**.
>
> 2. **Our Solution (AOEPT):** After identifying the bottleneck, a natural solution is to **inject global modality priors** associated with the missing modalities into the prompt. To further account for variation across samples, AOEPT realizes this in a **controlled and adaptive manner**. Specifically, for a sample $X = (x_1, * )$, AOEPT constructs prompts as:
> 	$$ P = I\left(C(x_1), M_2\right),$$
> 	where $M_2$ is the Modal-Contextualized Prompts (MCPs) derived from the global information of modality $\mathcal{X}_2$, and $I(\cdot)$ is the instance-aware instantiation (IAI) module.
>
> 	This design works in two steps: (1) **MCPs** provide a lightweight modality prior shared across samples; (2) **IAI module** further adapts MCPs to the current sample, activating the most relevant missing-modality information.
>
> 3. **Empirical Result**: In fact, we evaluated the mitigation of IMR in
> 	1. **Sec. 4.4 (Ablation Study):** Variant w/o MCP shows suboptimal performance.
> 	2. **Sec. 4.7 (NMI Analysis):** AOEPT achieves significantly higher NM2I scores than the competitive baseline.
> 	3. **Sec. 4.8 (Modality Information Scaling):** AOEPT benefits from increased available modality information which baselines can not, showing that it can escape the constraint of the bottleneck.
>
> > Q2: The robustness of AOEPT under test distribution shifts and low-resource settings.
>
> 1. First, existing modality-missing learning does not explicitly target **distribution shift robustness**. Nevertheless, our experiments already cover such scenarios. Particularly, the HateMemes dataset [1] involves an inherent distribution shift, as its test split is constructed with **manually controlled hate category proportions**. And our AOEPT achieves the superior results.
>
> 2. To further address your concerns, we introduce a cross-dataset setting [2], where the MT is trained on MOSI and evaluated on SIMS. We compare AOEPT with a strong baseline SyP, and the results in different missing type below show the robustness of AOEPT.
>
> |F1|A-T|A-V|T-V|V|T|A|
> |-|-|-|-|-|-|-|
> |SyP|42.04|37.82|41.76|40.69|31.89|39.96|
> |AOEPT|46.90|43.33|46.26|43.50|38.17|41.52|
>
> For **low-resource settings**, we provided evaluations in Appendix H.1. AOEPT outperforms baselines even under extreme conditions (i.e. 90% missing rates across training and inference).
>
> > Q3: Comparison with stronger prior/memory-augmented baselines.
>
> Indeed, we **have included** strong baselines augmented with retrieval and memory components (RAGPT, MemPrompt), and AOEPT outperforms them with a more efficient design.
>
> To further address this concern, we include a latest SOTA memory-based baseline, **REDEEM** [3]. As shown below, AOEPT outperforms it.
>
> |MM-IMDb|Text(F1-M)|F1-S|Image(F1-M)|F1-S|Both(F1-M)|F1-S|
> |-|-|-|-|-|-|-|
> |REDEEM|50.17|59.41|52.29|60.24|50.81|57.96|
> |AOEPT|51.50|59.69|54.86|61.06|53.31|59.92|
>
> ## Reference
>
> [1] The Hateful Memes Challenge, NeurIPS'20.
>
> [2] Smoothing the Shift: Towards Stable Test-Time Adaptation under Complex Multimodal Noises, ICLR'25.
>
> [3] REDEEMing Modality Information Loss: Retrieval-Guided Conditional Generation for Severely Modality Missing Learning, KDD'25.

---

### Decision · Program_Chairs · 2026-04-30

**Decision:**

Accept (regular)

**Comment:**

The submission is technically sound and well-supported by extensive experiments across multiple architectures (CLIP, ViLT, MulT) and diverse datasets. The review process highlighted a debate regarding the conceptual significance of the "IMR bottleneck." While one reviewer expressed concern that the empirical gains are incremental relative to the strong claims of a "critical bottleneck," the majority of the reviewers were convinced by the authors' technical evidence and diagnostic analyses.

Overall, the paper provides a well-executed and architecturally principled approach to a mature problem in multimodal learning. The introduction of a diagnostic metric (NM$^2$I) and the strong empirical performance make this a valuable contribution to the community.